# Efficient Learning with Sine-Activated Low-Rank Matrices

**Yiping Ji** [*]
Australian Institute for Machine Learning
University of Adelaide
DATA61, CSIRO

**Hemanth Saratchandran**[*]
Australian Institute for Machine Learning
University of Adelaide

**Cameron Gordon**
Australian Institute for Machine Learning
University of Adelaide

**Zeyu Zhang**
Australian National University

**Simon Lucey**
Australian Institute for Machine Learning
University of Adelaide

## Abstract

Low-rank decomposition has emerged as a vital tool for enhancing parameter efficiency in neural network architectures, gaining traction across diverse applications in machine learning. These techniques significantly lower the number of parameters, striking a balance between compactness and performance. However, a common challenge has been the compromise between parameter efficiency and the accuracy of the model, where reduced parameters often lead to diminished accuracy compared to their full-rank counterparts. In this work, we propose a novel theoretical framework that integrates a sinusoidal function within the low-rank decomposition. This approach not only preserves the benefits of the parameter efficiency of low-rank methods but also increases the decomposition's rank, thereby enhancing model performance. Our method proves to be a plug-in enhancement for existing low-rank methods, as evidenced by its successful application in Vision Transformers (ViT), Large Language Models (LLMs), Neural Radiance Fields (NeRF) and 3D shape modelling. The code is publicly available at https://yipingji.github.io/sine_activated_PEL/.

## 1 Introduction

In the last few years, large-scale machine learning models have shown remarkable capabilities across various domains, achieving groundbreaking results in tasks related to computer vision and natural language processing (Vaswani et al., 2017; Dosovitskiy et al., 2021). However, these models come with a significant drawback: their training necessitates an extensive memory footprint. This challenge has spurred the demand for more compact, parameter-efficient architectures. A prominent solution that has emerged is the use of low-rank techniques (Hu et al., 2022; Chen et al., 2024; Liu et al., 2024; Kopiczko et al., 2024), which involve substituting the large, dense matrices in large scale models with smaller, low-rank matrices. This substitution not only simplifies the models but also shifts the computational complexity from quadratic to linear, making a significant impact on efficiency. In the context of high-capacity models like Vision Transformers (ViTs) and Large Language Models (LLMs) that utilize millions to billions of parameters, transitioning from dense to low-rank matrices can result in considerable cost savings. Nonetheless, adopting low-rank architectures does introduce a trade-off, as they typically do not achieve the same level of accuracy as their full-rank counterparts, presenting a balance between parameter efficiency and model performance.

---

[*]Equal contribution. Correspondence to Yiping Ji <yiping.ji@adelaide.edu.au> and Hemanth Saratchandran <hemanth.saratchandran@adelaide.edu.au>.

In this paper, we tackle the challenge of balancing parameter efficiency and model performance by introducing a novel technique that enhances the representational capacity of low-rank methods. Our approach builds on the insight that augmenting low-rank matrices with high-frequency sinusoidal functions can increase their rank without adding parameters. We provide a theoretical framework that explains how this modulation increases rank and demonstrate how incorporating this non-linearity into low-rank decompositions enables compact architectures that preserve efficiency while achieving good accuracy across various machine learning tasks.

We direct the reader's attention to figure 1, which showcases our method across various machine learning applications. Our comparisons with standard low-rank methods consistently demonstrate superior performance across these diverse tasks.

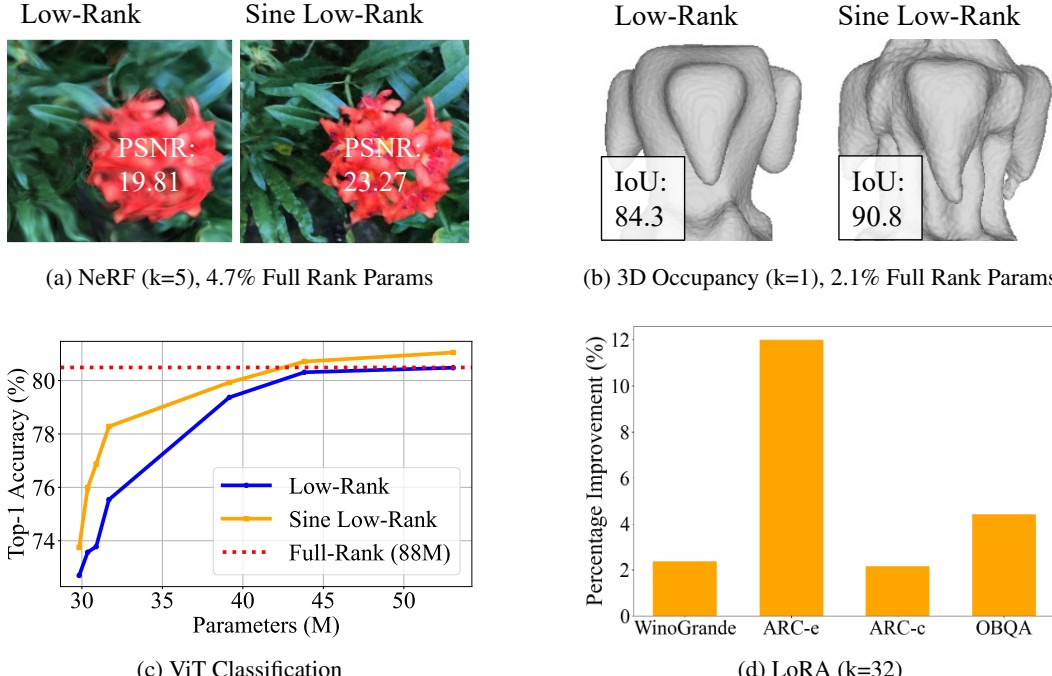

Figure 1: Applying a drop-in sine-activation increases the rank of low-rank matrix methods, leading to improved parameter efficiency and performance on a variety of tasks including: a) NeRF, b) 3D Occupancy, c) ViT image classification, and d) Fine-tuning Large Language Models (LoRA).

Our approach's advantages are corroborated across a range of machine learning applications, including low rank methods for ViT (He et al., 2022b), LLMs (Hu et al., 2022; Liu et al., 2024), NeRF for novel view synthesis (Mildenhall et al., 2020), and 3D shape modeling via Binary Occupancy Fields (Mescheder et al., 2019). Across the board, our approach not only matches the parameter savings offered by low-rank methods but also results in an improvement in accuracy, showing its broad applicability and superior performance among diverse machine learning tasks. The main contributions of our paper are:

1. A theoretical framework demonstrating that applying sinusoidal non-linearities to low-rank matrices can effectively increase their rank without introducing additional parameters.

2. Demonstrating that our theoretical framework leads to a drop-in component that can be applied to various low-rank architectures, resulting in improved accuracy while maintaining computational and parameter efficiency.

3. A comprehensive validation of our method across a range of diverse machine learning tasks, including computer vision, 3D shape modeling, and natural language processing.

## 2 RELATED WORK

### 2.1 LOW-RANK DECOMPOSITION:

Low-rank decomposition stands as a crucial method across disciplines such as information theory, optimization, and machine learning, providing an approach to reduce memory costs (Strang, 2019; Xinwei et al., 2023). Notably, (Candès et al., 2011) uncovered that matrices can precisely separate low-rank and sparse components through convex programming, linking to matrix completion and recovery. Expanding its application, (Yu et al., 2017) devised a low-rank learning framework for convolutional neural networks, enhancing compression while maintaining accuracy. (Sharma et al., 2023) found that performance improvements in large language models could be achieved by eliminating higher-order weight matrix components without extra parameters or data. In neural radiance fields, (Tang et al., 2022) introduced a rank-residual learning strategy for optimal low-rank approximations, facilitating model size adjustments. Additional contributions include (Shi & Guillemot, 2023) with rank-constrained distillation, (Chen et al., 2022) applying vector-matrix decomposition, and (Schwarz et al., 2023) using soft-gated low-rank decompositions for compression.

### 2.2 PARAMETER-EFFICIENT LEARNING:

Parameter-efficient learning is an important research area in deep learning, merging various techniques to enhance model adaptability with minimal resource demands (Menghani, 2023). Techniques like parameter-efficient fine-tuning (PEFT) allow pretrained models to adjust to new tasks efficiently, addressing the challenges of fine-tuning large models due to high hardware and storage costs. Among these, Visual Prompt Tuning (VPT) stands out for its minimal parameter alteration—less than 1%—in the input space, effectively refining large Transformer models while keeping the core architecture unchanged (Jia et al., 2022). Similarly, BitFit offers a sparse-finetuning approach, tweaking only the model's bias terms for cost-effective adaptations (Zaken et al., 2022). Moreover, LoRA introduces a low-rank adaptation that maintains model quality without additional inference latency or altering input sequence lengths, by embedding trainable rank matrices within the Transformer layers (Hu et al., 2022). Recent studies also combine LoRA with other efficiency strategies like quantization, pruning, and random projections for further model compression (Dettmers et al., 2024; Li et al., 2024; Zhang et al., 2024; Kopiczko et al., 2024; Liu et al., 2024).

## 3 METHODOLOGY

We introduce our main technique that we term a sine-activated low-rank approximation. The purpose of this technique is to increase the rank of a low-rank matrix without adding any extra parameters.

### 3.1 PRELIMINARIES

#### 3.1.1 FEED-FORWARD LAYER

Our technique is defined for feed-forward layers of a neural architecture. In this section, we fix the notation for such layers following Prince (2023). We express a feed-forward layer as:

$$\mathbf{y} = \mathbf{W}\mathbf{x} + \mathbf{b} \tag{1}$$

where $\mathbf{W} \in \mathbb{R}^{m \times n}$ is a dense weight matrix, $\mathbf{b} \in \mathbb{R}^{m \times 1}$ is the bias of the layer, and $\mathbf{x}$ is the input from the previous layer. The output $\mathbf{y}$ is often activated by a non-linearity $\sigma$ producing $\sigma(\mathbf{y})$. The weight matrix $\mathbf{W}$ and bias $\mathbf{b}$ are trainable parameters of the layer. In contemporary deep learning models, the feed-forward layers' weight matrices, $\mathbf{W}$, are often large and dense yielding a high rank matrix. While the high-rank property of the weight matrix helps in representing complex signals, it significantly adds to the overall parameter count within the network yielding the need for a trade-off between the rank of the weight matrix and overall architecture capacity.

#### 3.1.2 LOW-RANK DECOMPOSITION

A full-rank weight matrix $\mathbf{W}$ can be replaced by low-rank matrices $\mathbf{U}\mathbf{V}^T$, such that $\mathbf{W} = \mathbf{U}\mathbf{V}^T$, where $\mathbf{U} \in \mathbb{R}^{m \times k}$, $\mathbf{V} \in \mathbb{R}^{n \times k}$ and $k \ll min(m, n)$.

$$\mathbf{y} = \mathbf{W}\mathbf{x} + \mathbf{b} = (\mathbf{U}\mathbf{V}^T)\mathbf{x} + \mathbf{b} \tag{2}$$

This is the most common way to reduce the parameter count in a feed-forward layer. During the training process, this method performs optimization on $\mathbf{U}$ and $\mathbf{V}$ alternatively. Low-rank multiplication then reduces the learnable parameter count and memory footprint from $O(mn)$ to $O(k \cdot (m+n))$. Although $\mathbf{U}\mathbf{V}^T$ has the same matrix shape as the full-rank matrix $\mathbf{W}$, the rank of $\mathbf{U}\mathbf{V}^T$ is constrained and $rank(\mathbf{U}\mathbf{V}^T) \leq k$. Thus while we have significantly decreased the number of trainable weights in such a layer, we have paid the price by obtaining a matrix of much smaller rank. In the next section, we address this trade-off by developing a technique that can raise back the rank of a low-rank decomposition while keeping its low parameter count.

### 3.2 THEORETICAL FRAMEWORK

#### 3.2.1 NON-LINEAR LOW-RANK DECOMPOSITION

We introduce a non-linearity transformation into low-rank matrices as follows

$$\mathbf{y} = \frac{\phi(\omega \cdot \mathbf{U}\mathbf{V}^T)}{g}\mathbf{x} + \mathbf{b} \tag{3}$$

where $\phi(\cdot)$ is the non-linearity function, $\omega$ is a non-learnable frequency parameter, and $g$ is a non-learnable parameter to control the gain of the transformation. Our theoretical work will show that $\phi(\omega x) = sin(\omega x)$ is an optimal choice to increase the rank of $\mathbf{U}\mathbf{V}^T$.

#### 3.2.2 MAIN RESULT

In this section, we provide a theoretical framework that clearly shows how to increase the rank of a low-rank decomposition using a non-linearity without adding any parameters. We will show that if we choose the non-linearity, in the decomposition defined in section 3.2.1, to be a sine function then provided the frequency $\omega$ is chosen high enough, the rank of the matrix $\phi(\omega \cdot \mathbf{U}\mathbf{V}^T)$ will be larger than that of $\mathbf{U}\mathbf{V}^T$. The proofs of the theorems are given in the appendix A.1.

To begin with, we fix $\omega > 0$ and let $\sin(\omega \cdot \mathbf{A})$ denote the matrix obtained from a fixed $m \times n$ matrix $\mathbf{A}$ by applying the function $\sin(\omega \cdot \mathbf{x})$ component-wise to $\mathbf{A}$. Assuming $\mathbf{A} \neq 0$ we define $A_{\min}^0$ as:

$$A_{\min}^0 = \min_{i,j \ s.t. A_{ij} \neq 0} |A_{ij}|. \tag{4}$$

Note that such a quantity is well defined precisely because $\mathbf{A}$ has a finite number of entries and all such entries cannot be zero from the assumption that $\mathbf{A} \neq 0$.

The following theorem relates the rank of $\sin(\omega \cdot \mathbf{A})$ to the frequency $\omega$ and the quantity $A_{\min}^0$.

**Proposition 1.** *Fix an $m \times n$ matrix $\mathbf{A}$ s.t. $\mathbf{A} \neq 0$. Then*

$$Rank(\sin(\omega \cdot \mathbf{A})) \geq \omega \left( \frac{A_{\min}^0}{\left\| \sqrt{|\mathbf{A}|} \right\|_{op}} \right)^2 \quad if \quad 0 \leq \omega \leq \frac{\pi}{3A_{\min}^0}. \tag{5}$$

Proposition 1 shows that if we modulate the matrix $\sin(\omega \cdot \mathbf{A})$ by increasing $\omega > 0$ then the rank of the matrix $\sin(\omega \cdot \mathbf{A})$ can be increased provided $\omega < \frac{\pi}{3A_{\min}^0}$. We can apply proposition 1 to the context of a low-rank decomposition as defined in section 3.1.2. Given a low-rank decomposition $\mathbf{U}\mathbf{V}^T$ with $\mathbf{U} \in \mathbb{R}^{m \times k}$ and $\mathbf{V} \in \mathbb{R}^{n \times k}$ with $k \ll \min\{m, n\}$ the following theorem shows how we can increase the rank of the decomposition by applying a $\sin(\omega \cdot)$ function.

**Theorem 1.** *Let $\mathbf{U} \in \mathbb{R}^{m \times k}$ and $\mathbf{V} \in \mathbb{R}^{n \times k}$ with $k \ll \min\{m, n\}$. Assume both $\mathbf{U}$ and $\mathbf{V}$ are initialized according to a uniform distribution $\mathcal{U}(-1/N, 1/N)$ where $N > k$. Then there exists an $\omega_0$ such that the matrix $\sin(\omega \cdot \mathbf{A})$ will satisfy the inequality*

$$Rank(\sin(\omega \cdot \mathbf{U}\mathbf{V}^T)) > Rank(\mathbf{U}\mathbf{V}^T) \tag{6}$$

*provided $\omega \geq \omega_0$.*

Theorem 1 also holds for the case we initialize $\mathbf{U}$ and $\mathbf{V}$ by a normal distribution of variance $N$.

Weight matrices within feed-forward layers are typically initialized using a distribution that is contingent upon the layer's neuron count. When considering low-rank decompositions characterized by matrices $\mathbf{U} \in \mathbb{R}^{m \times k}$ and $\mathbf{V}^T \in \mathbb{R}^{k \times n}$, where $k \ll \min\{m, n\}$, the variance of this initialization distribution is influenced by $m$ and $n$. These dimensions are significantly larger than $k$, ensuring that the condition specified in theorem 1 — that $N > k$ — is always met, making this theorem especially relevant for low-rank decompositions in feed-forward layers. For example the most common initialization schemes such as Kaiming (He et al., 2015) and Xavier (Glorot & Bengio, 2010) satisfy the requirements of our theorem.

Theorem 1 offers a viable strategy for maintaining a high-rank characteristic in feed-forward layers while simultaneously minimizing the parameter count. By introducing a sinusoidal non-linearity with a sufficiently high frequency $\omega$ into a low-rank decomposition, it's possible to increase the rank of the layer without altering the quantity of trainable parameters.

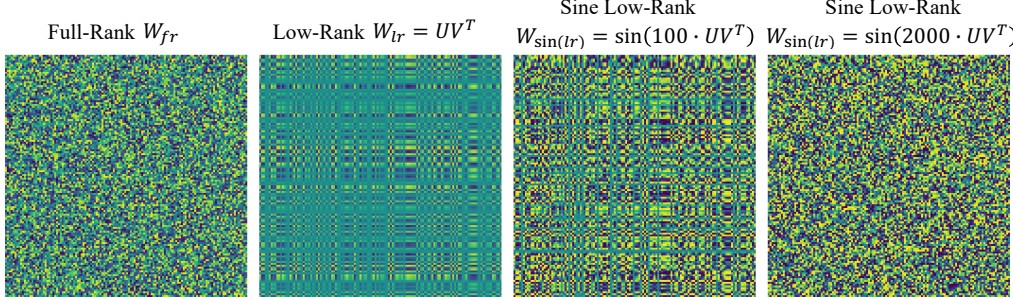

Figure 2: These figures display weight magnitudes for matrices with dimension $128 \times 128$. The first figure shows a heatmap of a full-rank matrix initialized by Kaiming uniform, highlighting linear independence among rows. The second shows a low-rank matrix $\mathbf{W}_{lr} = \mathbf{U}\mathbf{V}^T \in \mathbb{R}^{128 \times 128}$, with $\mathbf{U}, \mathbf{V} \in \mathbb{R}^{128 \times 1}$ illustrating minimal linear independence. The final pair of figures reveal how applying a sine function element-wise, $\sin(\omega \cdot \mathbf{U}\mathbf{V}^T)$, with varying $\omega$, affects linear independence in low-rank matrices; specifically, $\omega = 100$ and $\omega = 2000$ progressively increase linear independence.

In figure 2 we give a visualization of our method in action. We consider a full-rank matrix, a low-rank matrix, and two sine-activated low-rank matrices with different frequencies. By visualizing the weight magnitudes in each matrix via a heatmap, we can clearly see how the sine-activated low-rank matrix increases rank and furthermore how increasing the frequency of the sine function increases the rank in accord with theorem 1.

Building upon equation 3, we explore the application of various non-linear functions to a low-rank decomposition, with a particular focus on the sine function. This choice is inspired by theorem 1, which theoretically demonstrates that applying a sine function effectively increases the matrix rank. In figure 3, we present a comparative analysis of the sine function against other common non-linear functions in machine learning, such as the sigmoid and ReLU. The results clearly show that the sine function increases the rank, making it an optimal non-linearity to apply to a low-rank decomposition.

Further, theorem 1 suggests that augmenting the frequency of the sine function applied to a low-rank decomposition contributes to a further increase in rank. To empirically validate this, we conducted experiments applying sine functions of various frequencies to a constant low-rank matrix. The outcomes, depicted in figure 3 (right), corroborate the theorem's prediction, showcasing a positive correlation between the frequency of the sine function and the resultant rank increase.

## 4 EXPERIMENTS

This section is dedicated to validating and analyzing the efficacy of our proposed low-rank methods across a spectrum of neural network architectures. To demonstrate the broad applicability and versatility of our approach, we examine its performance in three distinct contemporary applications.

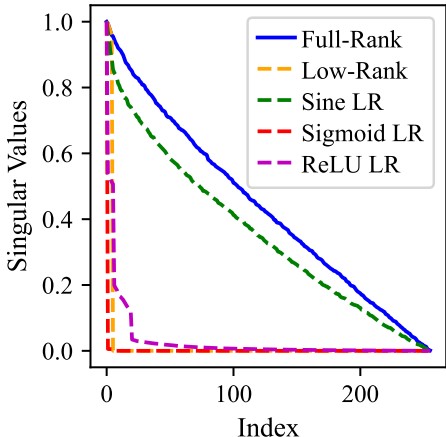 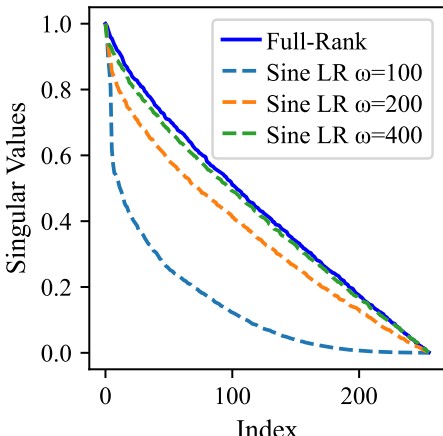

Figure 3: In this figure we depict the singular value spectrum of a Kaiming uniform initialized matrix $\mathbf{W}_{\text{fr}} \in \mathbb{R}^{256 \times 256}$ and a low-rank $k = 5$ approximation matrix $\mathbf{W}_{\text{lr}} = \mathbf{U}\mathbf{V}^T$. All singular values are normalized to 1. Left: the spectral advantages of applying a non-linear function $\phi(\omega \cdot \mathbf{U}\mathbf{V}^T)$ where $\omega$ is a hyper-parameter. Here we see the natural advantages of the sine function such that $\phi(\mathbf{x}) = \sin(\omega \cdot \mathbf{x})$. Right: manipulating $\omega$ within the sine function changes these spectral properties.

Specifically, we explore its integration into the fine-tuning of large language models through LoRA (Hu et al., 2022), the pretraining of ViT (Dosovitskiy et al., 2021), the reconstruction of scenes using NeRF (Mildenhall et al., 2020), and 3D shape modeling (Mescheder et al., 2019). This collectively underscores our model's adaptability to a diverse array of low-rank frameworks, highlighting its potential to significantly impact various domains within the field of computer vision.

## 4.1 LARGE LANGUAGE MODEL

LoRA is a highly effective strategy for finetuning large pretrained models, as described in (Hu et al., 2022). LoRA targets the adaptation of pretrained weight matrices $\mathbf{W}_0 \in \mathbb{R}^{m \times n}$ by limiting updates to a low-rank representation, expressed as $\mathbf{W}_0\mathbf{x} + \Delta\mathbf{W}\mathbf{x} = \mathbf{W}_0\mathbf{x} + \mathbf{U}\mathbf{V}^T\mathbf{x}$, where $\mathbf{U} \in \mathbb{R}^{m \times k}$ and $\mathbf{V} \in \mathbb{R}^{n \times k}$ with the rank $k \ll min\{m, n\}$. This method does not introduce additional inference latency or necessitate reducing the input sequence length, thus preserving the model quality. We conduct thorough experiments to evaluate the performance of our novel approach, termed sine LoRA, against the standard LoRA framework, demonstrating the effectiveness of our method.

**Dataset.** We evaluate the natural language understanding (NLU) task performance on the **RoBERTa V3** base model (Reimers & Gurevych, 2019). Specifically, we adopt the widely recognized GLUE benchmark (Wang et al., 2018), including CoLA (Warstadt et al., 2018), MRPC (Dolan & Brockett, 2005), QQP, STS-B(Cer et al., 2017), MNLI (Williams et al., 2018), QNLI (Rajpurkar et al., 2016), and RTE (Dagan et al., 2006; Haim et al., 2006; Giampiccolo et al., 2007; Bentivogli et al., 2009). Furthermore, we evaluate sine LoRA by fine-tuning large scale language models **LLaMA 3-8B** on commonsense reasoning tasks, which includes BoolQ (Clark et al., 2019), PIQA (Bisk et al., 2019), SIQA (Sap et al., 2019), HellaSwag (HS) (Zellers et al., 2019), WinoGrande (WG) (Sakaguchi et al., 2021), ARC-c, ARC-e (Clark et al., 2018) and OBQA (Mihaylov et al., 2018).

**Setting.** In the Transformer architecture, there are four weight matrices in the self-attention module $(\mathbf{W}_q, \mathbf{W}_k, \mathbf{W}_v, \mathbf{W}_o)$ and two in the MLP module$(\mathbf{W}_{\text{up}}, \mathbf{W}_{\text{down}})$. To evaluate RoBERTA V3, we follow up the LoRA architecture and implement low-rank adaptation only on $\mathbf{W}_q$ and $\mathbf{W}_v$. We study the performance of LoRA and sine LoRA in terms of different rank $k = 1, 2, 4, 8$. To evaluate LLaMA 3-8B, we implement low-rank adaptations on five modules $(\mathbf{W}_q, \mathbf{W}_k, \mathbf{W}_v, \mathbf{W}_{\text{up}}, \mathbf{W}_{\text{down}})$ with different rank $k = 4, 8, 16, 32$. For further implementation details see Appendix A.2.1.

**Results:** We replicated the experimental framework of naive LoRA to establish a baseline, and then evaluated our sine LoRA, as detailed in table 1 and table 2. Our results reveal that sine LoRA con-

Table 1: Performance and parameter count of the RoBERTa V3 model fine-tuned using the LoRA and sine LoRA methods across varying $k_{max}$ settings on the GLUE benchmark.

| Method | Params | COLA | MRPC | STSB | SST2 | RTE | QNLI | MNLI | QQP | Avg. | Δ |
|---|---|---|---|---|---|---|---|---|---|---|---|
| LoRA$_{k=1}$ | 36.9K | 66.31 | 90.15 | 90.15 | **94.70** | **78.80** | **93.06** | 88.18 | 87.61 | 85.63 | 0.32 ↑ |
| Sine LoRA$_{k=1}$ | | **67.99** | **90.44** | **90.85** | 94.79 | 78.05 | 92.76 | **88.35** | **87.90** | **85.95** | |
| LoRA$_{k=2}$ | 73.7K | 68.38 | 89.42 | 89.19 | **95.02** | 78.27 | **93.32** | **89.15** | 88.57 | 85.99 | 0.45 ↑ |
| Sine LoRA$_{k=2}$ | | **68.93** | **90.79** | **90.94** | 94.81 | **79.10** | 93.29 | 88.26 | **88.70** | **86.44** | |
| LoRA$_{k=4}$ | 147.5K | 68.56 | 89.69 | 88.79 | 95.23 | 80.39 | 93.34 | **89.78** | 88.70 | 86.41 | 0.73 ↑ |
| Sine LoRA$_{k=4}$ | | **68.93** | **90.86** | **90.87** | **95.25** | **82.00** | **93.53** | 89.68 | **89.18** | **87.14** | |
| LoRA$_{k=8}$ | 294.9K | **68.62** | 89.82 | 89.50 | **95.25** | 80.37 | 93.56 | **89.86** | 88.83 | 86.57 | 0.42 ↑ |
| Sine LoRA$_{k=8}$ | | 68.54 | **90.22** | **90.85** | 95.11 | **81.82** | **93.58** | 89.69 | **89.38** | **86.99** | |

Table 2: Performance and parameter count of the LLaMA 3-8B model fine-tuned using the LoRA and sine LoRA methods across varying $k_{max}$ settings on the commonsense reasoning benchmark.

| Method | Params | BoolQ | PIQA | SIMA | HS | WG | ARC-e | ARC-c | OBQA | Avg. | Δ |
|---|---|---|---|---|---|---|---|---|---|---|---|
| LoRA$_{k=4}$ | 7.1M | **73.58** | 86.29 | **79.99** | **94.92** | 79.95 | 63.91 | 78.7 | 83 | 80.04 | 3.57 ↑ |
| Sine LoRA$_{k=4}$ | | 72.69 | **87.38** | 79.32 | 94.39 | **85.32** | **75.01** | **88.64** | **86.2** | **83.61** | |
| LoRA$_{k=8}$ | 14.2M | 72.97 | **87.43** | 78.81 | 72.18 | 85.80 | **77.47** | **88.38** | 83.20 | 80.79 | 2.87 ↑ |
| Sine LoRA$_{k=8}$ | | **73.42** | 86.51 | **80.3** | **94.16** | **85.87** | 76.36 | 88.05 | **84.6** | **83.66** | |
| LoRA$_{k=16}$ | 28.3M | 73.57 | 85.58 | 79.27 | 93.97 | **85.71** | 75.42 | 86.44 | 83.2 | 82.9 | 2.45 ↑ |
| Sine LoRA$_{k=16}$ | | **73.7** | **87.65** | **80.76** | **94.93** | 84.45 | **79.1** | **89.77** | **84.4** | **84.35** | |
| LoRA$_{k=32}$ | 56.6M | 70.64 | 86.13 | 78.25 | 91.48 | 83.19 | 69.71 | 85.73 | 81.4 | 80.82 | 2.74 ↑ |
| Sine LoRA$_{k=32}$ | | **72.42** | **86.51** | **79.78** | **93.96** | **85.16** | **78.07** | **87.58** | **85** | **83.56** | |

sistently surpasses the performance of the standard LoRA at different rank levels ($k$), highlighting the effectiveness of the sine function in enhancing the representation capabilities of low-rank matrices. Notably, sine LoRA at $k = 4$ not only exceeds LoRA's performance at $k = 8$ by 0.57 but also halves the parameter count, illustrating significant efficiency and parameter savings. Surprisingly, with the LLaMA 3-8B model, our method with rank 4 already outperforms LoRA with all higher ranks, achieving 83.61 compared to 82.9.

**Analysis.** Within the LoRA framework, featuring a low-rank multiplication component $\Delta \mathbf{W} = \mathbf{UV^T}$, we enhance this low-rank component with a sine function and assess the efficacy of our method. This adaptation amplifies the update significance due to the 'intrinsic rank' increase, facilitated by the sine-activation. Consequently, our approach attains superior performance at reduced rank levels $k$, compared to LoRA, effectively decreasing the count of learnable parameters.

**Further results.** For comparisons to the recent DORA paper (Liu et al., 2024) see Appendix A.2.1.

## 4.2 PRETRAINING VISION TRANSFORMERS

Vision Transformers have risen to prominence as powerful models in the field of computer vision, demonstrating remarkable performance across a variety of tasks. When pretrained on large-scale datasets such as ImageNet-21K and JFT-300M, ViTs serve as robust foundational architectures, particularly excelling in feature extraction tasks (Deng et al., 2009; Sun et al., 2017). A critical observation regarding the architecture of ViTs is that the two feed-forward layers in each block dedicated to channel mixing contribute to nearly 66% of the total model parameter count. In light of this, focused experiments on these specific layers have been conducted to rigorously assess the effectiveness of our proposed method, facilitating a direct comparison with the baseline model.

**Experimental setup.** We trained the ViT-Small and ViT-Base models from scratch, utilizing the CIFAR-100 and ImageNet-1k datasets, respectively, to establish our baseline performance metrics (Deng et al., 2009; Krizhevsky, 2012). The ViT-Small model, characterized by its two MLP layers with input/output dimensions of 384 and hidden dimensions of 1536, was modified by replacing the full-rank weight matrices with low-rank matrices across a range of ranks ($k$). Similarly, the ViT-Base model, which features two MLP layers with input/output dimensions of 768 and hidden dimensions of 3072, underwent a parallel modification, where its full-rank weight matrices were

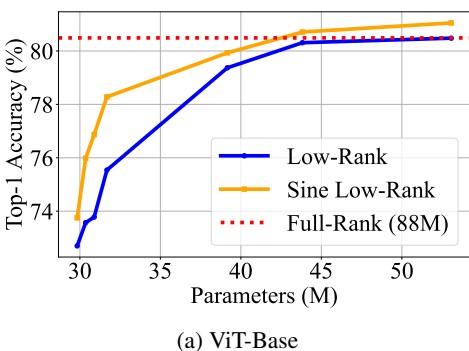
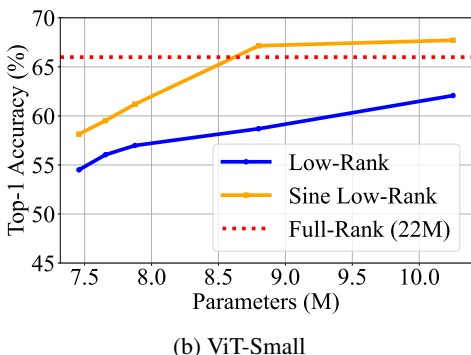

(a) ViT-Base  (b) ViT-Small

Figure 4: Low Rank ViT classification performance. Use of the sine-activation improves performance of the low-rank models, and even enables improvement relative to the Full Rank model.

substituted with low-rank matrices for a range of ranks. For the training of the ViT-Base model, we follow the training methodology described in Masked Autoencoders (MAE) (He et al., 2022a), implementing a batch size of 1024. This structured approach allows us to rigorously evaluate the impact of introducing low-rank matrices to these model architectures. For further implementation details see Appendix A.2.2.

**Results.** Figure 4 shows the outcomes of training ViT models from scratch on the ImageNet-1k and CIFAR100 datasets, respectively. These findings are compared with those of conventional baseline training of ViT models, which demonstrate that employing aggressive low-rank levels ($k$) compromises accuracy. Remarkably, the ViT-Base model, even when operating at a rank of 250 with only 50% of its parameters in comparison to the baseline, attains the performance metrics of the baseline on the ImageNet-1k dataset, albeit at the cost of increased training loss. The use of sine low-rank matrices consistently yields substantial improvements in test accuracy across all examined rank levels for both datasets. This suggests that the sine function significantly bolsters the representational capacity of low-rank weight matrices, as suggested by the theory in section 3.2.

**Analysis.** Large models, such as ViT-Base, with an excessively large number of parameters, are prone to overfitting, where they perform well on training data but poorly on unseen data, especially when trained on relatively 'small' datasets like ImageNet-1k (Xu et al., 2024). Low-rank learning techniques can help in designing models that generalize better to new data by encouraging the model to learn more compact and generalizable representations to reduce overfitting. Additionally, while ViT architectures often underperform on smaller datasets, this method introduces a novel approach for efficiently training ViT models using small data collections. For frequency ablations in the rank 1 case, see Appendix A.2.2.

**Further results: ConvNeXt.** In order to show that our method works on convolutional only architectures we implemented sine low-rank on ConvNeXt (Liu et al., 2022), a leading convolutional architecture for image classification. For implementation details and results see Appendix A.2.3.

### 4.3 NERF

Neural Radiance Fields (NeRFs) represent 3D scene signals by utilizing a set of 2D sparse images (Mildenhall et al., 2020). The 3D reconstruction is obtained by a forward pass $f_\theta(x, y, z, \theta, \phi)$, involving position $(x, y, z)$ and viewing direction $(\theta, \phi)$. We evaluate our methods by training a NeRF model on the standard benchmarks LLFF dataset, which consists of 8 real-world scenes captured by hand-held cameras (Mildenhall et al., 2019). To evaluate our method on NeRF we substitute each fully dense layer with low-rank decomposition and use a range of rank levels ($k$).

**Results.** Table 3 provides results on the LLFF dataset (Mildenhall et al., 2019; 2020). We report the peak signal-to-noise ratio (PSNR) with the compression rate representing the percentage of parameters used in comparison to the parameter count of the Full Rank NeRF model. Employing low-rank matrices in NeRF learning reduces parameter count while significantly enhancing compression. However, performance dips with very low-rank levels ($k$), where models capture minimal informa-

Table 3: Quantitative results for NeRF evaluated on the LLFF dataset.

| | Fern | Flower | Fortress | Horn | Leaves | Orchids | Room | Trex | Average | Δ | Compression Rate |
|---|---|---|---|---|---|---|---|---|---|---|---|
| | | | | | PSNR↑ | | | | | | |
| Full-Rank | 26.38 | 27.54 | 30.93 | 28.20 | 21.79 | 21.33 | 30.96 | 27.68 | 26.85 | - | 100% |
| Low-Rank$_{k=1}$ | 15.03 | 14.60 | 14.74 | 13.66 | 12.89 | 12.50 | 15.04 | 13.54 | 14.00 | | |
| Sine Low-Rank$_{k=1}$ | **20.77** | **20.14** | **24.13** | **19.00** | **15.92** | **16.25** | **25.53** | **16.42** | **19.77** | 5.77 ↑ | 1.3% |
| Low-Rank$_{k=5}$ | 20.64 | 19.81 | 24.90 | 20.40 | 15.74 | 16.07 | 22.74 | 19.79 | 20.01 | | |
| Sine Low-Rank$_{k=5}$ | **23.50** | **23.27** | **26.78** | **23.99** | **18.49** | **18.90** | **27.05** | **22.96** | **23.11** | 3.10 ↑ | 4.7% |
| Low-Rank$_{k=10}$ | 22.83 | 22.18 | 25.96 | 22.76 | 17.36 | 18.12 | 26.12 | 21.69 | 22.12 | | |
| Sine Low-Rank$_{k=10}$ | **24.56** | **24.61** | **28.01** | **25.39** | **19.62** | **20.02** | **28.70** | **24.21** | **24.39** | 2.27 ↑ | 8.7% |
| Low-Rank$_{k=30}$ | 24.48 | 24.68 | 28.10 | 25.54 | 19.36 | 20.04 | 38.92 | 24.24 | 24.42 | | |
| Sine Low-Rank$_{k=30}$ | **25.71** | **26.01** | **29.46** | **27.16** | **20.95** | **21.17** | **30.18** | **26.27** | **25.86** | 1.45 ↑ | 24.6% |
| Low-Rank$_{k=60}$ | 25.26 | 26.16 | 29.50 | 26.74 | 20.39 | 20.85 | 30.00 | 25.81 | 25.59 | | |
| Sine Low-Rank$_{k=60}$ | **26.09** | **26.70** | **29.75** | **27.78** | **21.56** | **21.37** | **30.54** | **27.16** | **26.36** | 0.77 ↑ | 48.6% |

tion. Our methods, nevertheless, substantially elevate performance. For instance, with $k = 1$, our sine low-rank approach yields an average PSNR of 19.77, outperforming the naive low-rank by 5.77 and achieving a compression rate of merely 1.3%. Even at a 48% compression rate, it surpasses the basic low-rank model by 0.77 PSNR, narrowly trailing the baseline by just 0.49 PSNR, as shown in Figure 5b. Our rate-distortion analysis, applying Akima interpolation for Bjøntegaard Delta calculation, reveals a BD-Rate of $-64.72\%$ and BD-PSNR of 2.72dB, signifying marked improvements in compression efficiency (Bjøntegaard, 2001; Herglotz et al., 2022). Visualization for $k = 1$ results are shown in Figure 5a and more results are shown in Figure 8 in the Appendix.

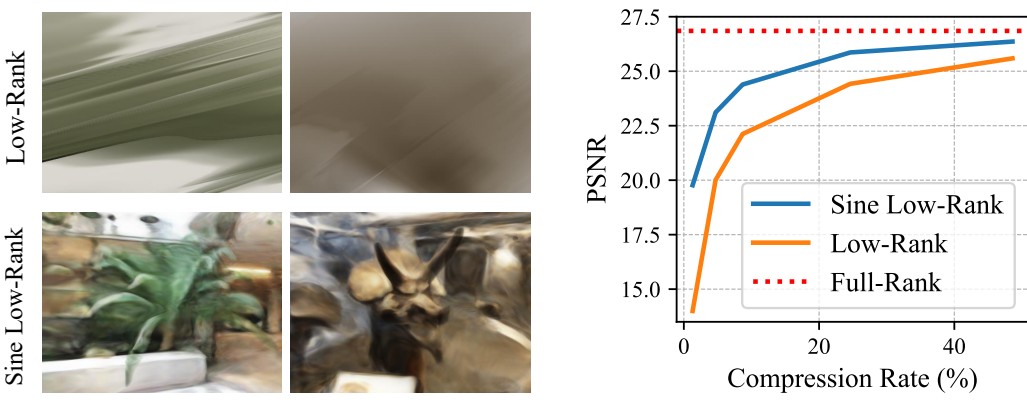

(a) Qualitative NeRF results for LLFF datasets ($k = 1$).      (b) Rate-Distortion curve (LLFF average).

Figure 5: (a) Using a non-transformed Low-Rank model leads to a complete loss of signal at extreme (rank $k = 1$). In contrast, applying a sine-activation function is able to reconstruct details even at 1.3% of the Full-Rank parameters. (b) The Sine Low-Rank NeRF models show significant improvements across the rate-distortion curve relative to the Low-Rank models.

**Analysis.** NeRF models fit entire 3D scenes, and a high training PSNR leads to a high testing PSNR (Mildenhall et al., 2020). Employing structured weight matrices could result in a drop in performance due to the inherent constraints imposed by their structural design. Increasing the matrices' rank enhances their memorization abilities significantly, especially when using a very low $k$. Starting from a low frequency, there is a rapid and consistent increase in PSNR. Consequently, as we elevate the rank level $k$, our results gradually align with the baseline NeRFs, which serve as the upper bound. For frequency ablations in the case of rank 1 and rank 5 see Appendix A.2.4.

### 4.4 3D SHAPE MODELING

For this experiment, we evaluate performance on binary occupancy field reconstruction, which involves determining whether a given coordinate is occupied (Mescheder et al., 2019). Following (Saragadam et al., 2023), we sampled over a $512 \times 512 \times 512$ grid with each voxel within the volume assigned a 1, and voxels outside the volume assigned a 0. We use the Thai Statue, Dragon and Lucy instance from the Stanford Scanning Repository.[1] We evaluate intersection over union (IoU) for the occupancy volumes. We used a coordinate-based MLP that includes two hidden layers, each with a width of 256 neurons, and employed the Gaussian activation function. The full-rank model achieves an accuracy of 97 (IoU). Figure 6 shows the 3D mesh representation of the Thai Statue, visualized using the low-rank method and the sine low-rank method for $k = 1, 2, 5$. Applying the sine function to the low-rank matrix resulted in a significant enhancement and more precise shape delineation. Results on Dragon and Lucy are given in Table 13 in Appendix A.2.5. A frequency ablation in the rank 1 case is given in Appendix A.2.5.

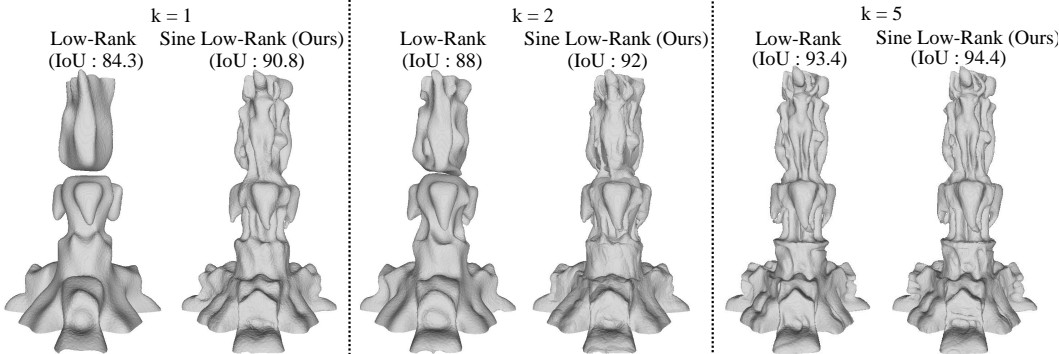

Figure 6: Binary occupancy field reconstruction on the Thai Statue. Note that without a sine function, the low-rank model is unable reconstruct any finer details for the $k = 1$ case; however, even at that level the sine low-rank model is able to reconstruct fine structural details of the statue, including the trunks of the elephants. The $k = 1$, $k = 2$ and $k = 5$ model utilizes only 2.1%, 2.9% and 5.2%, respectively, of the parameters of the full-rank model.

## 5 LIMITATIONS

Our exploration into sine low-rank matrices illuminates their promising capabilities, yet it also has a limitation: notably, while these matrices can reach rank levels comparable to their full-rank counterparts upon the application of a sine function, their accuracy falls short. This highlights an ongoing challenge in finding the optimal balance between the need for sufficient parameterization to ensure high accuracy and the preferable rank of matrices. Overparameterization is widely recognized in the literature as vital for deep learning models to achieve strong generalization and memorization. Moving forward, developing strategies that not only increase rank but also clearly define the necessary degree of overparameterization will be crucial for creating cost-effective deep learning architectures, presenting an intriguing avenue for future research.

## 6 CONCLUSION

In this work we have demonstrated that applying a sinusoidal non-linearity improves the accuracy of low-rank approximations by increasing their rank. While simple, this method is highly applicable to parameter-constrained models such as LoRA, as it improves approximation without adding capacity, by overcoming representation limits of the matrix rank. We have fully justified this approach from theoretical first principles. When applied as a drop-in component we showed that this method leads to surprisingly large improvements across a range of tasks involving low-rank models, including language tasks, image classification, neural radiance fields and 3D shape modelling.

---

[1]Available at https://graphics.stanford.edu/data/3Dscanrep/

ACKNOWLEDGEMENTS

Hemanth Saratchandran and Simon Lucey acknowledge support from Commonwealth Bank of Australia through the CommBank Centre for Foundational AI Research. This funding was essential for the completion of the research described in this publication.

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

# A  APPENDIX

## A.1  THEORETICAL FRAMEWORK

In this section we give the proof of proposition 1 and theorem 1 from section 3.2 of the paper.

We recall from section 3.2 the following notation: For fixed $\omega > 0$, let $\sin(\omega \cdot \mathbf{A})$ denote the matrix obtained from a fixed $m \times n$ matrix $\mathbf{A}$ by applying the function $\sin(\omega \cdot \mathbf{x})$ component-wise to $\mathbf{A}$. Assuming $\mathbf{A} \neq 0$ we define $A_{\min}^0$ as:

$$A_{\min}^0 = \min_{i,j \, s.t. A_{ij \neq 0}} |A_{ij}|. \tag{7}$$

Note that such a quantity is well defined precisely because $\mathbf{A}$ has a finite number of entries and all such entries cannot be zero from the assumption that $\mathbf{A} \neq 0$.

Before we give the proof of proposition 1 from section 3.2 of the main paper, we will prove two lemmas.

**Lemma 1.** *For a fixed $m \times n$ matrix $\mathbf{A}$. We have*

$$||sin(\omega \mathbf{A})||_F^2 \geq \omega^2 (A_{\min}^0) \; if \, 0 < \omega < \frac{\pi}{3A_{\min}^0} \tag{8}$$

*where $A_{\min}^0$ is defined as follows:*

$$A_{\min}^0 = \min_{i,j \, s.t. A_{ij \neq 0}} |A_{ij}| \tag{9}$$

*for $1 \leq i \leq m$ and $1 \leq j \leq n$.*

*Proof.* Observe by definition of the Frobenius norm that

$$||sin(\omega \mathbf{A})||_F^2 = \sum_{i=1}^{m} \sum_{j=1}^{n} sin(\omega \mathbf{A}_{ij})^2. \tag{10}$$

We then find that

$$||sin(\omega \mathbf{A})||_F^2 \geq sin(\omega A_{\min}^0)^2. \tag{11}$$

The goal is to now find a lower bound on $sin(\omega A_{\min}^0)^2$. In order to do this consider the function $f(\omega) = sin(\omega x) - \frac{\omega x}{2}$, where $x \in \mathbb{R}$ is fixed and positive.

Differentiating this function we have

$$f'(\boldsymbol{\omega}) = x cos(\boldsymbol{\omega} x) - \frac{x}{2}. \tag{12}$$

To find a critical point we solve the equation $f'(\boldsymbol{\omega}) = 0$ to find

$$cos(\boldsymbol{\omega} x) = \frac{1}{2}. \tag{13}$$

We see that equation 13 has the solution $\omega x = \frac{\pi}{3}$. In order to check what type of critical point $\boldsymbol{\omega} x = \frac{\pi}{3}$ we need to look at $f''(\boldsymbol{\omega})$

$$f''(\boldsymbol{\omega}) = -x^2 sin(\boldsymbol{\omega} x) < 0 \tag{14}$$

when $\omega = \frac{\pi}{3x}$ implying that the critical point $\omega = \frac{\pi}{3x}$ is a maximum point.

Observe that $f(0) = 0$ it thus follows that $f(\boldsymbol{\omega}) \geq 0$ on the interval $[0, \frac{\pi}{3x}]$.

Applying this to the function $sin(\boldsymbol{\omega} A_{\min}^0)$ we obtain that

$$sin(\boldsymbol{\omega} A_{\min}^0) \geq \frac{\boldsymbol{\omega} A_{\min}^0}{2} \; if \, 0 \leq \boldsymbol{\omega} \leq \frac{\pi}{3A_{\min}^0}. \tag{15}$$

Substituting the lower bound in equation 15 into equation 11 we obtain the proposition. $\square$

The next lemma establishes an upper bound on the operator norm of $sin(\omega A)$. We remind the reader that the operator norm of $A$ is denoted by $||A||_{op}$ and is defined by

$$||A||_{op} = \sup_{||x||_2=1} ||Ax||_2 \tag{16}$$

where $x$ is a vector and $||\cdot||_2$ represents the vector 2-norm. It can be shown that the operator norm of $A$ is also given by the largest singular value of $A$ see (Strang, 2019).

**Lemma 2.** *Let $A$ be a fixed $m \times n$ matrix. Then*

$$||sin(\omega \mathbf{A})||_{op}^2 \leq \left|\left|\sqrt{\omega}\sqrt{|\mathbf{A}|}\right|\right|_{op}^2 \tag{17}$$

*where $\sqrt{|\mathbf{A}|}$ denotes the matrix obtained from $A$ by taking the absolute value and then square root component wise.*

*Proof.* By definition we have

$$||sin(\omega \mathbf{A})||_{op}^2 = \sup_{||x||_2=1} ||sin(\omega \mathbf{A})x||_2^2 \tag{18}$$

where $||\cdot||_2$ denotes the 2-norm of a vector.

For any fixed unit vector $x$ we will show how to upper bound the quantity $||sin(\omega \mathbf{A})x||_2^2$. In order to do this we will use the fact that for $x \geq 0$, we have the bound $sin(x) \leq \sqrt{|x|}$.

$$||sin(\omega \mathbf{A})x||_2^2 = \sum_{i=1}^{m}\left(\sum_{j=1}^{n} sin(\omega A_{ij})x_j\right)^2 \tag{19}$$

$$\leq \sum_{i=1}^{m}\left(\sum_{j=1}^{n} \sqrt{\omega}\sqrt{|\mathbf{A}_{ij}|}x_j\right)^2 \tag{20}$$

$$= ||(\sqrt{\omega})\left(\sqrt{|\mathbf{A}|}\right)x||_2^2. \tag{21}$$

It follows that

$$\sup_{||x||=1} ||sin(\omega \mathbf{A})x||_2^2 \leq \sup_{||x||=1} ||\sqrt{\omega}\sqrt{|\mathbf{A}|}x||_2^2 \tag{22}$$

which implies

$$||sin(\omega \mathbf{A})||_{op}^2 \leq \left|\left|\sqrt{\omega}\sqrt{|\mathbf{A}|}\right|\right|_{op}^2. \tag{23}$$

$\square$

We can now give the proof of proposition 1 from section 3.2 of the main paper. In order to do so we will need the definition of the stable rank of a matrix. Assume $\mathbf{A}$ is a non-zero $m \times n$ matrix. We define the stable rank of $\mathbf{A}$ by

$$SR(A) := \frac{||\mathbf{A}||_F^2}{||\mathbf{A}||_{op}^2}. \tag{24}$$

It is easy to see from the definition that the stable rank is continuous, unlike the rank, and is bounded above by the rank

$$SR(\mathbf{A}) \leq Rank(\mathbf{A}). \tag{25}$$

**Remark 1.** *We observe that lemmas 1 and 2 give the main reasons why we chose a sine function as the non-linearty to apply on a weight matrix $\mathbf{A}$. The periodic nature of a sine function that can be controlled by a frequency parameter $\omega > 0$ is what allows us to obtain a proof of lemmas 1 and 2. When we give a proof of Thm. 1 we will see that these two lemmas are crucial.*

*of proposition 3.1 from section 3.2 of main paper.* Observe that from equation 25 it suffices to prove the lower bound on $SR(\mathbf{A})$. This is immediate from lemma 1 and lemma 2. $\square$

*Proof of theorem 1 from section 3.2 of main paper.* From the assumption of the theorem 1 we have that $N >> k$. Further, we are assuming that both $\mathbf{U}$ and $\mathbf{V}$ have entries sampled from $\mathcal{U}(-1/N, 1/N)$. This means if we let $\mathbf{A} = \mathbf{U}\mathbf{V}^T$, then there exists a $C > 0$ such that

$$A_{\min}^0 = \frac{C}{N^2}. \tag{26}$$

Furthermore, observe that

$$||\sqrt{|\mathbf{A}|}||_{op} \leq ||\sqrt{|\mathbf{A}|}||_F \leq ||\mathbf{A}||_F \tag{27}$$

which implies

$$\omega\left(\frac{A_{\min}^0}{||\sqrt{|\mathbf{A}|}||_{op}}\right)^2 \geq \omega\left(\frac{C}{N^4}\right)\left(\frac{N^4}{mn}\right) = \omega(\frac{C}{mn}). \tag{28}$$

Now observe that from proposition 1 in section 3.2 from the main paper we have that

$$Rank(sin(\omega \mathbf{A})) \geq \omega \frac{A_{\min}^0}{||\sqrt{|\mathbf{A}|}||_{op}} \tag{29}$$

if $0 \leq \omega \leq \frac{\pi}{3A_{\min}^0}$. We can rewrite this last condition to say that equation 29 holds if $0 \leq \omega \leq \frac{\pi N^2}{3}$. In particular, by using equation 28 we find that there exists $\omega_0$ within the interval $0 \leq \omega_0 \leq \frac{\pi N^2}{3}$

$$Rank(sin(\omega_0 \mathbf{A})) \geq \omega_0 \frac{A_{\min}^0}{||\sqrt{|\mathbf{A}|}||_{op}} \geq k \geq Rank(\mathbf{A}). \tag{30}$$

This completes the proof. □

**Remark 2.** *Observe that if $\omega$ is very small, then $\sin(\omega x) \approx \omega x$ and thus applying a $\sin$ simply scales the matrix by $\omega$ which cannot change rank. It is only when $\omega$ is sufficiently large that we see that the rank increases. If $\omega = 0$ then applying $\sin$ with frequency $\omega$ to the matrix produces the zero matrix which has zero rank and in general will thus have rank less than $A$.*

**Remark 3.** *In general low-rank matrices inherently have fewer degrees of freedom compared to high-rank matrices, which limits their representational capacity. For complex datasets, we hypothesize that high-rank weight matrices within a neural model provide additional degrees of freedom, enabling the model to capture and learn key features more effectively from the input data. Our empirical results on all tasks seem to validate this hypothesis.*

## A.2 EXPERIMENTS

For the experiments we observed that the gain factor $g$ in Equation (3) should be chosen analogously to how weights are initialized in (He et al., 2015). In particular we chose $g = \sqrt{n}$, where $n$ was the number of rows of the weight matrix. During backpropagation the frequency parameter $\omega$ scales the gradients which can cause gradient exploding. To mitigate this we found the choice of $g = \sqrt{n}$ worked best. We also point out that for the experiments there is no principled way to set $\omega$. Therefore, we will obtain $\omega$ by treating it as a hyperparameter and tuning it according to best results.

### A.2.1 FINE TUNING LARGE LANGUAGE MODELS

**Implementation details for Roberta V3** We followed the settings in (Hu et al., 2022) and (Ding et al., 2023). In the Transformer architecture, there are four weight matrices in the self-attention module ($\mathbf{W}_q, \mathbf{W}_k, \mathbf{W}_v, \mathbf{W}_o$) and two in the MLP module($\mathbf{W}_{up}, \mathbf{W}_{down}$). To evaluate RoBERTA V3, we follow up the LoRA architecture and implement low-rank adaptation only on $\mathbf{W}_q$ and $\mathbf{W}_v$. We study the performance of **LoRA** and **sine LoRA** in terms of different rank $k = 1, 2, 4, 8$. For sine LoRA, we use frequency = 200 across all the ranks. We use different learning rate and epoch for different datasets as shown in Table 4.

**Implementation details for Llama3-8B** We followed the settings in (Liu et al., 2024). In the Transformer architecture, there are four weight matrices in the self-attention module ($\mathbf{W}_q, \mathbf{W}_k, \mathbf{W}_v, \mathbf{W}_o$) and two in the MLP module($\mathbf{W}_{up}, \mathbf{W}_{down}$). To evaluate Llama3-8B, we implement low-rank adaptation only on $\mathbf{W}_q, \mathbf{W}_k, \mathbf{W}_v, \mathbf{W}_{up}$, and $\mathbf{W}_{down}$. We study the performance of **LoRA** and **sine LoRA** for different rank $k = 4, 8, 16, 32$ and configurations are as shown in 5.

| Dataset | lr | epoch |
|---------|------|-------|
| CoLA | 8e-5 | 20 |
| SST-2 | 1e-4 | 10 |
| MRPC | 1e-4 | 20 |
| QQP | 3e-4 | 10 |
| STS-B | 1e-4 | 20 |
| MNLI | 3e-4 | 10 |
| QNLI | 3e-4 | 10 |
| RTE | 1.2e-3 | 50 |

Table 4: Learning rate and epoch for each dataset for the Roberta V3 model.

| rank | frequency | lr | epoch |
|------|-----------|------|-------|
| 4 | 800 | 1e-4 | 3 |
| 8 | 600 | 1e-4 | 3 |
| 16 | 200 | 1e-4 | 3 |
| 32 | 200 | 1e-4 | 3 |

Table 5: Sine LoRA. Frequency, learning rate and epochs for the Llama3-8B model.

**DoRA**  In order to compare our method to the current state-of-the-art we apply our sine-LoRA method to Weight-Decomposed Low-Rank Adaptation (DoRA) (Liu et al., 2024). DoRA decomposes pre-trained weights $\mathbf{W} \in \mathbb{R}^{m \times n}$ into two components: a magnitude vector $q \in \mathbb{R}^{1 \times n}$ and a direction matrix $\mathbf{D} \in \mathbb{R}^{m \times n}$, normalized by the column vector-wise norm $\| \cdot \|_c$ such that each column remains a unit vector.

DoRA decompose a low-rank change in the direction matrix into the decomposition $\mathbf{\Delta D} = \mathbf{U V^T}$, where $\mathbf{U} \in \mathbb{R}^{m \times k}$ and $\mathbf{V} \in \mathbb{R}^{n \times k}$. To implement our **sine-DoRA**, we apply the sine function to this decomposition as per Equation (3). We evaluate the performance of DoRA and sine-DoRA in terms of different rank $k = 8, 16, 32$ in Table 7 and configurations are shown in Table 8. We implement $k = 8$ directly as this is not a setting used in (Liu et al., 2024), and compare against reported results for $k = 16$ and $k = 32$. By introducing our methods on top of DoRA, our model (Sine-DoRA $k = 8$) achieve state-of-the-art results while utilizing only 25% of the parameters required by DoRA($k = 32$).

**Computational cost.**  Finetuning Llama3-8B takes roughly 6 hours using LoRA, 7 hours using Sine LoRA, 11 hours using DoRA, and 11 hours using Sine DoRA using a NVIDIA H100 GPU with 96GB of memory. Training memory cost is shown in Table 6.

### A.2.2 VISION TRANSFORMERS

**Implementation details for ViT-Small on CIFAR100:**  The ViT-Small model, characterized by its two MLP layers with input/output dimensions of 384 and hidden dimensions of 1536, was modified by replacing the full-rank weight matrices with low-rank matrices across a range of ranks ($k$). We use learning rate 1e-3, batch size 512 and train for 200 epochs. Choices of frequency for different ranks are shown in Table 9.

Table 6: Training memory (GB) cost for LoRA, Sine LoRA, DoRA, Sine Dora on finetuning Llama3-8B as reported by Nvidia-SMI.

| Method | Rank 8 | Rank 16 | Rank 32 |
|--------|--------|---------|---------|
| LoRA | 54.9 | 55.2 | 55.3 |
| Sine LoRA | 72.0 | 72.0 | 72.2 |
| DoRA | 75.6 | 75.9 | 76.0 |
| Sine DoRA | 90.4 | 90.7 | 90.8 |

Table 7: Performance and parameter count of the LLaMA 3-8B model fine-tuned using the DoRA and sine DoRA methods across varying $k_{max}$ settings on the commonsense reasoning benchmark. * Results reported in the paper.

| Method | Params | BoolQ | PIQA | SIQA | HS | WG | ARC-e | ARC-c | OBQA | Avg. | Δ |
|---|---|---|---|---|---|---|---|---|---|---|---|
| DoRA$_{k=8}$ | 14.9M | 73.2 | 87.7 | 79.9 | 94.7 | 84.5 | 89.3 | 78.0 | 83.2 | 83.8 | 1.4 ↑ |
| Sine DoRA$_{k=8}$ | | **73.9** | **89.0** | **81.0** | **95.3** | **86.1** | **90.1** | **79.0** | **87.0** | **85.2** | |
| DoRA*$_{k=16}$ | 29.1M | 74.5 | 88.8 | 80.3 | **95.5** | 84.7 | **90.1** | 79.1 | 87.2 | 85.0 | 0.3 ↑ |
| Sine DoRA$_{k=16}$ | | **75.1** | **89.0** | **81.0** | 95.3 | **86.1** | 90.0 | **79.3** | 86.2 | **85.3** | |
| DoRA*$_{k=32}$ | 57.4M | 74.6 | **89.3** | 79.9 | 95.5 | 85.6 | **90.5** | **80.4** | **85.8** | 85.2 | 0.1 ↑ |
| Sine DoRA$_{k=32}$ | | **75.8** | **89.3** | **80.3** | **95.9** | **86.1** | 90.2 | 79.4 | 85.4 | **85.3** | |

Table 8: Sine DoRA. Frequency, learning rate and epochs for the Llama3-8B model.

| Rank | Frequency | Learning Rate | Epoch |
|---|---|---|---|
| 8 | 300 | 6e-5 | 3 |
| 16 | 150 | 6e-5 | 3 |
| 32 | 100 | 6e-5 | 3 |

**Implementation details for ViT-Base on ImageNet-1k:** We followed the settings in (He et al., 2022a). We use batch size of 1024, learning rate of 3e-4 and we train for 300 epochs. Choices of frequency for different ranks are shown in Table 9.

Table 9: Frequencies used for different ranks for ViT-Base and ViT-Small.

| | rank | 1 | 5 | 10 | 30 | 60 | 100 | 150 | 250 |
|---|---|---|---|---|---|---|---|---|---|
| ViT-small (CIFAR100) | $\omega$ | 500 | 500 | 300 | 300 | 300 | - | - | - |
| ViT-Base (ImageNet-1k) | $\omega$ | 1000 | 800 | 800 | 400 | 200 | 200 | 150 | 150 |

**Ablation on frequencies:** In table 10, we examine the performance of our method on training the ViT-Small model from scratch on the CIFAR100 dataset using different frequencies, when $k = 1$.

Table 10: Top-1 Accuracy of ViT-Small ($k = 1$) on CIFAR100 with varying frequencies $\omega$.

| Frequency $\omega$ | 100 | 200 | 300 | 400 | 500 | 600 | 700 |
|---|---|---|---|---|---|---|---|
| **PSNR** | 55.0 | 55.8 | 56.8 | 58.0 | **58.1** | 57.6 | 57.5 |

### A.2.3 CONVNEXT ON CIFAR100

ConvNeXt is a family of convolutional neural networks (CNNs) models introduced in (Liu et al., 2022). These models are designed to modernize traditional CNNs architectures by incorporating design elements inspired by Vision Transformers (ViTs) to enhance performance in image recognition.

**Implementation details:** We employ our methods on ConvNeXt-Tiny model using CIFAR100 datasets and the Timm codebase. ConvNeXt-Tiny consists of 4 stages with block numbers [3, 3, 9, 3] and feature dimensions with [96, 192, 384, 768]. The majority of parameters (50%) in ConvNeXt are used in the last stage, therefore we apply a low rank decomposition only to the linear feature layer in this stage. We use a batch size of 512, learning rate of 5e-3, and we train for 150 epochs.

**Results:** In Table 11, we present our performance and configurations. We demonstrate that our method consistently outperform the naive low rank method, and even outperforms the baseline full-rank method (ConvNext-Tiny) with approximately 50% fewer parameters.

Table 11: Performance and compression rate of ConvNeXt-Tiny model trained on CIFAR100 datasets. We use frequency [400, 300, 300] for rank [1, 5, 20] respectively.

| Method | # Params | Acc % | Compression Rate % |
|---|---|---|---|
| ConvNeXt-Tiny | 27.9M | 62.3 | 100 |
| LR-ConvNeXt-Tiny$_{k=1}$
Sine LR-ConvNeXt-Tiny$_{k=1,\omega=400}$ | 13.8M | 59.5
**61.5** | 49.5 |
| LR-ConvNeXt-Tiny$_{k=5}$
Sine LR-ConvNeXt-Tiny$_{k=5,\omega=300}$ | 13.9M | 59.5
**62.0** | 50.0 |
| LR-ConvNeXt-Tiny$_{k=20}$
Sine LR-ConvNeXt-Tiny$_{k=20,\omega=300}$ | 14.2M | 62.1
**62.8** | 50.9 |

### A.2.4  NeRF

**Implementation details:**  We followed the settings in (Ramasinghe & Lucey, 2022). We use 8 fully connected layers–each with 256 neurons, a learning rate of 5e-4 and train for 500k iterations. We evaluate the performance of our method by experimenting with different ranks [1, 5, 10, 30, 60], corresponding to frequencies [1400, 800, 600, 400, 300] respectively.

**Results:**  The full qualitative results on NeRF are given in figure 8.

**Ablations:**  In figure 7, we illustrate the impact of varying frequency on PSNR for cases where k=1 (shown on the left) and k=5 (shown on the right).

**Computational cost:**  In Table 12, we present the training memory usage (MB) on NeRF experiments across different ranks.

Figure 7: Ablation NeRF results for the LLFF dataset. These two figures show PSNR of NeRF using different frequencies, when $k = 1$ (on the left) and $k = 5$ (on the right)

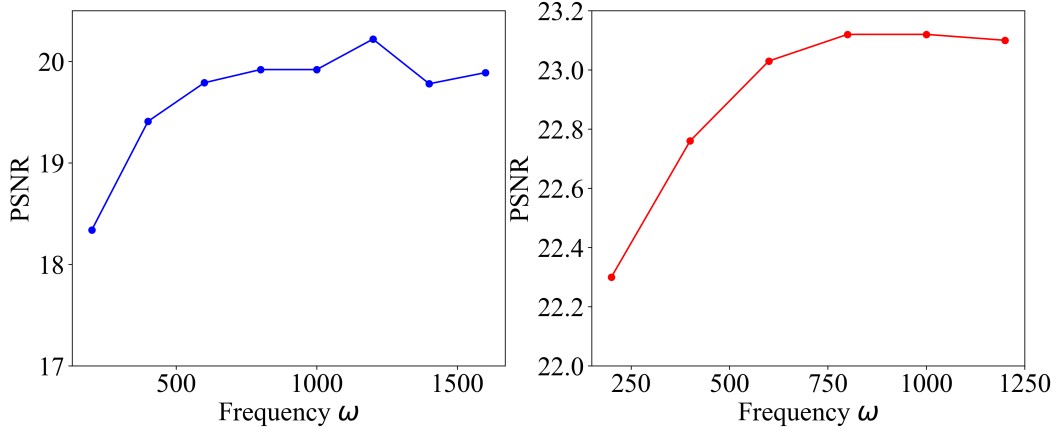

Table 12: Training memory usage(MB) on NeRF experiments

| Method | rank 1 | rank 5 | rank 10 | rank 30 | rank 60 |
|---|---|---|---|---|---|
| Low-Rank | 5620 | 5622 | 5622 | 5626 | 5630 |
| Sine Low-Rank | 5630 | 5630 | 5632 | 5634 | 5638 |

Figure 8: Qualitative NeRF results for the LLFF dataset (Mildenhall et al., 2019; 2020) using rank $k = 1$ and $k = 5$. Using a low-rank model leads to a complete loss of signal for $k = 1$, however, applying sine is able to reconstruct details even at the extreme low-rank case. At $k = 5$ the sine low-rank model is noticeably sharper and clearer than using the low-rank.

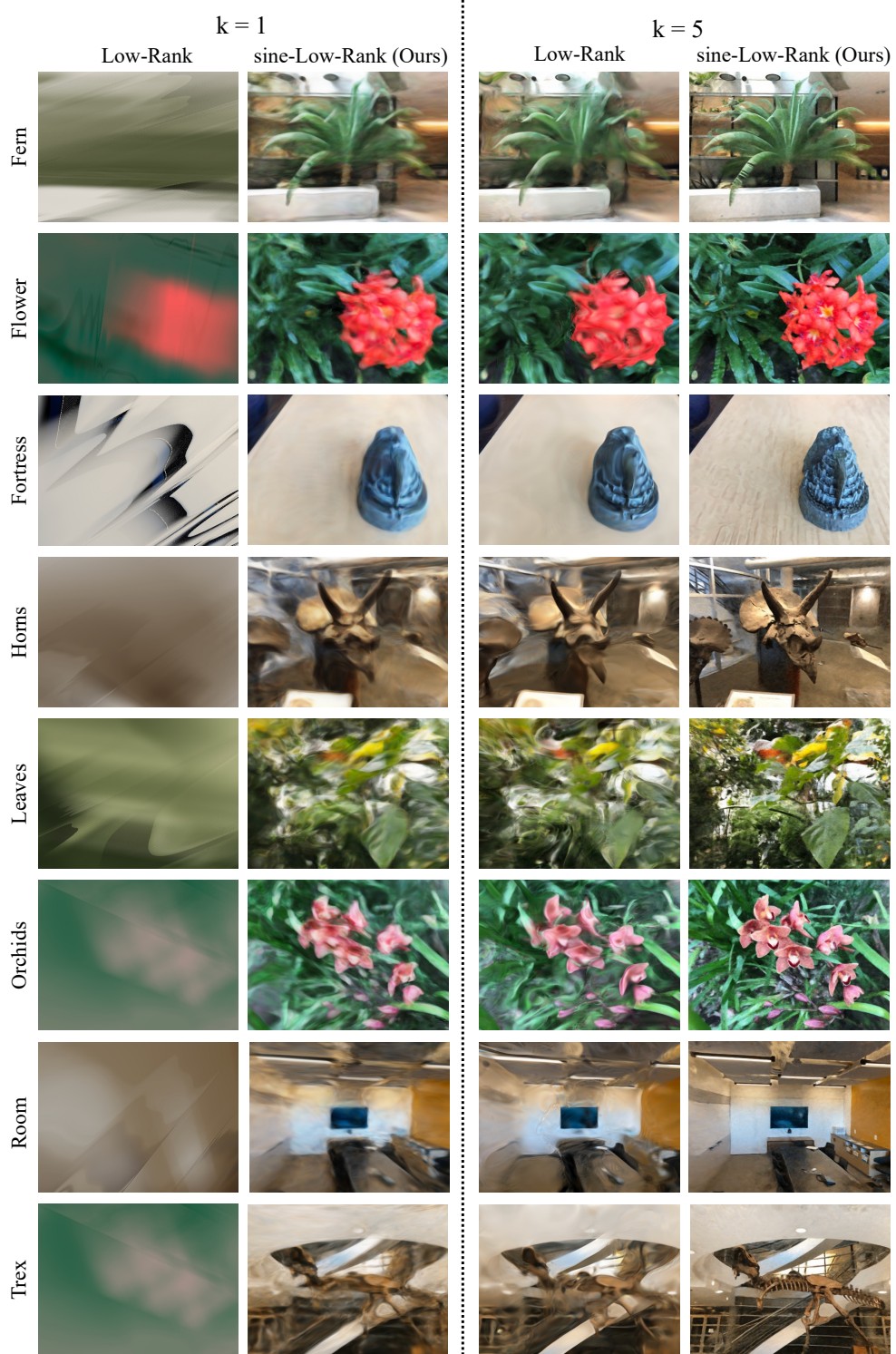

Figure 9: Ablation binary occupancy results for Thai Statue. This figure shows IoU accuracy of 3D shape modeling using different frequencies, when k=1.

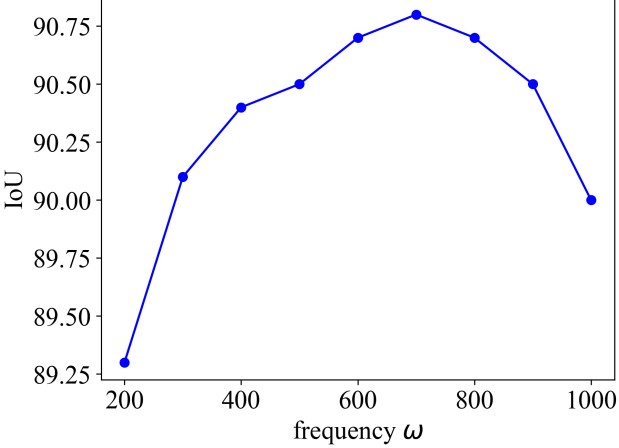

### A.2.5 3D SHAPE MODELLING

**Implementation details:** We use 2 fully connected layers–each with 256 neurons, a learning rate of 1e-3 and train for 200 epochs.

**Results:** Table 13 reports the Intersection over Union (IoU) and Compression Rate of the binary occupancy task using different rank levels (k). Our sine low-rank methods.

Table 13: This table illustrates Intersection over Union for 3D shape modeling (Thai Statue, Lucy and Dragon) across different rank levels ($k$). It also includes the compression rate, indicating the proportion of parameters utilized relative to the total parameter count of the baseline model, thereby detailing the parameter usage versus model performance at different levels of model complexity. We use frequency [200, 100, 50, 20] for ranks [1, 2, 5, 20], respectively

|  | # Params | IoU | | | Compression Rate |
|---|---|---|---|---|---|
|  |  | Thai | Lucy | Dragon |  |
| Full-Rank | 132K | 97.2 | 97.8 | 98.7 | 100% |
| Low-Rank$_{k=1}$ 
 Sine Low-Rank$_{k=1,\omega=200}$ | 2.8K | 84.3 
 **90.8** | 79.3 
 **90.7** | 90.4 
 **94.6** | 2.1% |
| Low-Rank$_{k=2}$ 
 Sine Low-Rank$_{k=2,\omega=100}$ | 3.8K | 88.0 
 **92.0** | 89.4 
 **93.2** | 90.9 
 **96.6** | 2.9% |
| Low-Rank$_{k=5}$ 
 Sine Low-Rank$_{k=5,\omega=50}$ | 6.9K | 93.4 
 **94.3** | 94.8 
 **95.3** | 96.9 
 **97.4** | 5.2% |
| Low-Rank$_{k=20}$ 
 Sine Low-Rank$_{k=20,\omega=20}$ | 22.8K | **95.4** 
 95.4 | 96.2 
 **96.3** | 98.0 
 **98.1** | 16.8% |

