# OpenReview forum: "Efficient Learning with Sine-Activated Low-Rank Matrices"
_ICLR.cc/2025/Conference — ICLR 2025 Poster_

### Official Review · Reviewer_EgEA · 2024-11-03

**Soundness:** 3
**Presentation:** 2
**Contribution:** 2
**Rating:** 6
**Confidence:** 4

**Summary:**

This paper introduces concise and elegant improvement method for Low Rank Adaption (LoRA) and reveals its underlying principle by rigorous formula derivation. In addition to fine-tuning large language model, this method can even also achieving competitive results when directly replacing original full-rank weight, which provides some inspiration for model compression. Although there are some parts that I think are slightly lacking in depth, I would suggest accepting it after the following issues are addressed.

**Strengths:**

This paper introduces concise and elegant improvement method for Low Rank Adaption (LoRA) and reveals its underlying principle by rigorous formula derivation. In addition to fine-tuning large language model, this method can even also achieving competitive results when directly replacing original full-rank weight, which provides some inspiration for model compression. Although there are some parts that I think are slightly lacking in depth, I would suggest accepting it after the following issues are addressed.

**Weaknesses:**

Advice & Question:
1. In formula 3, why the coefficient w and g are non-learnable ? If learnable, will it cause gradient problem? For the same model, has it been considered to provide different w and g for different linear layers? Can the recommended values for these two be provided?
2. Since other nonlinear functions are almost useless, why not directly give the sine function in Equation 3?
3. Regarding the right diagram in Figure 3, I don't think we can see a trend of the matrix rank increasing with the increase of frequency w. Because the rank of a matrix is only related to the number of nonzero singular values. So when w is 200 and 400, it seems that the ranks of the two are almost equal.
4. In the setting of section 4.1, why do not implement low-rank adaptations on all linear modules of LLM?
5. Does the rank of the decomposed matrix necessarily be less than that of the original matrix regardless of how the frequency is adjusted?

**Questions:**

Advice & Question:
1. In formula 3, why the coefficient w and g are non-learnable ? If learnable, will it cause gradient problem? For the same model, has it been considered to provide different w and g for different linear layers? Can the recommended values for these two be provided?
2. Since other nonlinear functions are almost useless, why not directly give the sine function in Equation 3?
3. Regarding the right diagram in Figure 3, I don't think we can see a trend of the matrix rank increasing with the increase of frequency w. Because the rank of a matrix is only related to the number of nonzero singular values. So when w is 200 and 400, it seems that the ranks of the two are almost equal.
4. In the setting of section 4.1, why do not implement low-rank adaptations on all linear modules of LLM?
5. Does the rank of the decomposed matrix necessarily be less than that of the original matrix regardless of how the frequency is adjusted?

---

> ### Author Response · Authors · 2024-11-21
> **Response by authors**
>
> We thank the reviewer for their review and greatly appreciate the time and effort they have put in reviewing our paper. Below we will answer each of the reviewers questions in detail. We have also made changes as requested by the reviewer and have uploaded an updated version of our paper.
>
> 1. **In formula 3, why the coefficient w and g are non-learnable ? If learnable, will it cause gradient problem? For the same model, has it been considered to provide different w and g for different linear layers? Can the recommended values for these two be provided?**
>
> In order for these coefficients to be learnable we found that initializing them the right way was hugely important. If we initialized them at zero we found training was affected and if we initialized them too large, we witnessed exploding gradients. For $g$ we followed the approach taken in the paper by He et al. [1]. We set $g$ to be $\sqrt{n}$ where $n$ is the number of rows of the weight matrix (i.e. the fan\_in in PyTorch). This was chosen for the following reason. During backpropagation the sin function has a frequency $\omega$ and when taking a gradient the frequency $\omega$ comes out the front scaling each gradient. So as to mitigate gradient explosion we found that taking $g = \sqrt{n}$ worked well. For the frequency unfortunately there is currently no principled way to choose $\omega$. Theorem 1 suggests that larger values of $w$ is better but it does not quantify how large. Therefore, for each experiment and each rank we treated $w$ as a hyperparameter and tuned it to what gave the best results. We have made a remark about this for the reader in the appendix, see remark 2 in the appendix.
>
> [1] Delving Deep into Rectifiers: Surpassing Human-Level Performance on ImageNet Classification by He et al. ICCV 2015.
>
> 2. **Since other nonlinear functions are almost useless, why not directly give the sine function in Equation 3?**
>
> We thank the reviewer for pointing this out and agree with their comment. Our main reason for including a general $\phi$ is to motivate the procedure as a general construction. We have added a sentence in line 171 saying that our work shows that sin is the best choice.
>
> 3. **Regarding the right diagram in Figure 3, I don't think we can see a trend of the matrix rank increasing with the increase of frequency w. Because the rank of a matrix is only related to the number of nonzero singular values. So when w is 200 and 400, it seems that the ranks of the two are almost equal.**
>
>  We understand the reviewers point with this figure. The purpose of figure 3 is to show that by applying a sin function with different frequencies we can move the spectrum to the right. For example if you look at the right figure in figure 3 you can see that the case of $w = 100$ there are few singular values that are almost zero but in the case of $w = 200$ these have become much larger showing that the numerical rank can be increased.
>
> 4. **In the setting of section 4.1, why do not implement low-rank adaptations on all linear modules of LLM?**
>
> For our implementation of low rank adapations on linear module we followed work in the literature.
> For RoBERTa the low rank adaptation is applied only on $W_q$ and $W_v$ as this is what was done in the original LoRA paper and we wanted to compare our method directly with LoRA. However, for LLaMa3-8B the low rank adaptation is applied on $W_q$, $W_k$,$W_v$, $W_{up}$ and $W_{down}$ as this is what was done in the recent DoRA paper [2] and we wanted to compare our method directly with DoRA. This gives two different situations of applying low rank adaptations and in both cases our method performs better. Results of our method on DoRA can be found in A.2.1 Table 7. As can be seen from that table sine DoRA outperforms DoRA at all rank levels.
>
>
> [2] Liu et al. DoRA: Weight-Decomposed Low-Rank Adaptation. ICML 2024
>
>
> 5. **Does the rank of the decomposed matrix necessarily be less than that of the original matrix regardless of how the frequency is adjusted?**
>
> If $\omega$ is very small, then $\sin(\omega x) \approx \omega x$ and thus applying a sin simply scales the matrix by $\omega$ which cannot change rank. It is only when $\omega$ is large that we see that the rank increases. If $\omega = 0$ then applying $sin$ with frequency $\omega$ to the matrix produces the zero matrix which has zero rank and in general will thus have rank less than $A$. We have included remark 2 in the appendix that explains this.

---

> > ### Author Response · Authors · 2024-11-26
> >
> > Further to our response to question 4 of the reviewer regarding the linear modules of the LLM. We draw the reviewers attention to experiments we conducted on state of the art DoRA [2], that applies our sine low-rank on the wider set of linear modules ($W_q$, $W_k$,$W_v$, $W_{up}$ and $W_{down}$) in DoRA.
> >
> > As can be seen from the table our sine DoRA outperforms DoRA at each rank level and rank 8 sine DoRA matches the performance of rank 32 DoRA.
> >
> > [2] Liu et al. DoRA: Weight-Decomposed Low-Rank Adaptation. ICML 2024
> >
> > **Performance and parameter count of the LLaMA 3-8B model fine-tuned using the DoRA and sine DoRA methods across varying ranks**
> >
> > | **Method**              | **Params** | **BoolQ** | **PIQA** | **SIQA** | **HS**  | **WG**  | **ARC-e** | **ARC-c** | **OBQA** | **Avg.** |
> > |--------------------------|------------|-----------|----------|----------|---------|---------|-----------|-----------|----------|----------|
> > | DoRA$_{k=8}$      | 14.9M      | 73.2      | 87.7     | 79.9     | 94.7    | 84.5    | 89.3      | 78.0      | 83.2     | 83.8     |
> > | Sine DoRA$_{k=8}$ | 14.9M      | **73.9**  | **89.0** | **81.0** | **95.3**| **86.1**| **90.1**  | **79.0**  | **87.0** | **85.2** |
> > | DoRA$_{k=16}$    | 29.1M      | 74.5      | 88.8     | 80.3     | **95.5**| 84.7    | **90.1**  | 79.1      | 87.2     | 85.0     |
> > | Sine DoRA$_{k=16}$| 29.1M      | **75.1**  | **89.0** | **81.0** | 95.3    | **86.1**| 90.0      | **79.3**  | **86.2** | **85.3** |
> > | DoRA$_{k=32}$    | 57.4M      | 74.6      | **89.3** | 79.9     | 95.5    | 85.6    | **90.5**  | **80.4**  | **85.8** | 85.2     |
> > | Sine DoRA$_{k=32}$| 57.4M      | **75.8**  | **89.3** | **80.3** | **95.9**| **86.1**| 90.2      | 79.4      | 85.4     | **85.3** |

---

> ### Author Response · Authors · 2024-11-29
>
> Dear reviewer EgEA, we thank you very much for your review. Before the discussion period ends, we would like to check if our response has addressed the reviewers concerns or if there are remaining issues we may clarify that would increase your confidence that this paper should be accepted?

---

### Official Review · Reviewer_HVQH · 2024-11-04

**Soundness:** 4
**Presentation:** 3
**Contribution:** 3
**Rating:** 8
**Confidence:** 3

**Summary:**

This paper addresses the critical need for compact and parameter-efficient architectures in machine learning. While low-rank techniques are a popular solution for reducing model complexity, the authors introduce high-frequency sinusoidal non-linearity functions to enhance the capabilities of low-rank matrices. The paper provides a theoretical framework demonstrating how this modulation increases the rank of matrices without adding extra parameters. This sine-based non-linearity is incorporated into low-rank decompositions, effectively balancing efficiency and performance improvements across various benchmarks, including Vision Transformers (ViT), Large Language Models (LLMs), Neural Radiance Fields (NeRF), and 3D shape modeling.

**Strengths:**

Task Motivation:
The paper tackles a highly relevant problem—parameter-efficient learning. While overparameterization is effective for generalization, practical deployment in industry necessitates efficient architectures for cost-effectiveness.

Simple Yet Effective Approach:
The introduction of a sine function to augment low-rank decompositions is straightforward but impactful. As shown in Figure 3, the Sine LR method maintains representational power comparable to other activation functions while improving efficiency. The effectiveness of this method is demonstrated across multiple tasks and backbones, including LoRA in LLMs (ROBERTaV3 and LLaMA 3-8B), ViT (ViT-base and ViT-small), NeRF (vanilla NeRF on LLFF), and 3D shape modeling (Thai Statue example).

Technical Soundness:
The theoretical analysis highlights how controlling the frequency coefficient "w" in the sine function affects the linear independence of the original matrix "W". Figure 2 illustrates this effect, showing that the sine-activated low-rank decomposition facilitates efficient training without increasing the number of model parameters.

**Weaknesses:**

Lack of Rationale Behind the Sine Function's Effectiveness: While the theoretical and experimental validations in Sections 3.2 and 4 establish the efficacy of the sine activation, the paper doesn't delve into why the sine function specifically enhances parameter-efficient training. I'm left wondering what inherent properties of the sine function drive these results. A clearer explanation of how the sine function contributes to increasing matrix rank or improving performance without adding parameters would significantly strengthen the argument.

Hyperparameter "w" Tuning:
The role of the frequency hyperparameter "w" isn't thoroughly addressed. Is there a need to fine-tune "w" for each task, or does it generally benefit from being larger? More importantly, how do we determine the optimal "w"? A systematic exploration or guidelines for selecting "w" across different tasks would be helpful, as the current approach seems to leave this critical aspect underexplored.


Applicability to CNN Architectures:
Although the paper demonstrates strong results on various architectures like ViT, LLMs, NeRF, and 3D shape modeling, it leaves a gap concerning CNNs. Given that CNNs are still widely used in practice, it would be valuable to explore whether the sine-activated low-rank method can be effectively applied to CNN architectures. A discussion or preliminary results in this area would broaden the paper’s applicability and relevance.

**Questions:**

Please refer the weakness section above.

---

> ### Author Response · Authors · 2024-11-21
> **Response by authors**
>
> We thank the reviewer for their review and greatly appreciate the time and effort they have put in reviewing our paper. Below we will answer each of the reviewers questions in detail. We have also made changes as requested by the reviewer and have uploaded an updated version of our paper.
>
> 1. **Lack of Rationale Behind the Sine Function's Effectiveness: While the theoretical and experimental validations in Sections 3.2 and 4 establish the efficacy of the sine activation, the paper doesn't delve into why the sine function specifically enhances parameter-efficient training.......**
>
>  Thank you very much for your question. Our main reason for using a sine function is that it has periodic behaviour around the origin that can be controlled by a frequency parameter. This allows us to show that for any matrix $\mathbf{A}$, that is not the zero matrix, there are frequencies $\omega > 0$ that allow us to lower bound the Frobenius norm of
> $\sin(\omega\mathbf{A})$ by a quantity that depends linearly on the frequency $\omega$ and upper bound the operator norm of
> $\sin(\omega\mathbf{A})$ by a quantity that depends sub-linearly on the frequency $\omega$. This is the content of lemmas 1 and 2 in appendix A.1 and are the two key reasons why we can increase the rank of $\mathbf{A}$ after applying a sine function with a large enough frequency. When choosing a non-linear function to increase rank we found that our theory implied the function should satisfy such properties. This is how we then came to the conclusion that a sine function would be a good choice.
> The proof of both lemmas uses the fact that sine is a periodic function whose frequency can be controlled by a positive parameter $\omega > 0$ which is what we found was critical. In general, other non-linearities like Sigmoid and ReLU do not satisfy such properties and this is why we found that they cannot be made to increase rank. We have added a remark in the appendix after the proof of lemma 2 that clearly explains this for the reader.
>
> 2. **Hyperparameter "w" Tuning: The role of the frequency hyperparameter "w" isn't thoroughly addressed. Is there a need to fine-tune "w" for each task, or does it generally benefit from being larger? More importantly, how do we determine the optimal "w"? A systematic exploration or guidelines for selecting "w" across different tasks would be helpful, as the current approach seems to leave this critical aspect underexplored.**
>
> Yes we treat the frequency $w$ as a hyperparameter and then do a search for the best parameters. In general the bigger $w$ is we found the better, as this also follows from Thm. 1, however the theorem does not quantify exactly how large we should take $w$ to be. Due to this we found we had to run a search to tune for the best $w$ for each experiment. It is our feeling that a principled approach to choosing $w$ for a given application is an extremely interesting problem that we hope to take up in future work. We have carried out ablations on the frequency in the appendix. Please see A.2.2 for the case of ViT-small, A.2.4 for the case of NeRF and A.2.5 for the case of 3D shape modelling.
>
> 3. **Applicability to CNN Architectures: Although the paper demonstrates strong results on various architectures like ViT, LLMs, NeRF, and 3D shape modeling, it leaves a gap concerning CNNs.....**
>
> We thank the reviewer for their suggestion. As per requested we ran our method on the ConvNeXt-Tiny architecture [1], a state of the art convolutional architecture, on the CIFAR100 dataset. The results are shown appendix A.2.3 Table 11. For the convenience of the reviewer we show the results in the table below. As can be seen, our sine low-rank method achieves better accuracy at all rank levels and at rank 20 outperform the original full rank ConvNeXt-Tiny architecture.
>
>
> **Performance of ConvNeXt-Tiny model trained on CIFAR100 datasets.**
> *We use frequency [400, 300, 300] for rank [1, 5, 20], respectively.*
>
> | **Method**                           | **# Params** | **Acc (%)** |
> |--------------------------------------|--------------|-------------|
> | ConvNeXt-Tiny                        | 27.9M        | 62.3        |
> | LR-ConvNeXt-Tiny$_{k=1}$       | 13.8M        | 59.5        |
> | Sine LR-ConvNeXt-Tiny$_{k=1, ω=400}$ |    13.8M          | **61.5**   |
> | LR-ConvNeXt-Tiny$_{k=5}$      | 13.9M        | 59.5        |
> | Sine LR-ConvNeXt-Tiny$_{k=5, ω=300}$ |    13.9M          | **62.0**   |
> | LR-ConvNeXt-Tiny$_{k=20}$     | 14.2M        | 62.1        |
> | Sine LR-ConvNeXt-Tiny$_{k=20, ω=300}$ |     14.2M         | **62.8**   |

---

> ### Comment · Reviewer_HVQH · 2024-11-27
>
> Thank you for your author rebuttal.
>
> The authors have provided a clear explanation of the rationale behind the effectiveness of the sine function (W1). While the need for further exploration of the hyperparameter $w$ remains, this can be addressed in future work. Additionally, the effectiveness of the proposed approach within the CNN architecture has been adequately demonstrated.
>
> I maintain my stance that this study should be accepted.

---

### Official Review · Reviewer_Gygt · 2024-11-04

**Soundness:** 3
**Presentation:** 3
**Contribution:** 3
**Rating:** 8
**Confidence:** 3

**Summary:**

This paper proposes the introduction of sinusoidal functions within LORAs to improve model performance across a range of tasks. The claim is to be able to balance parameter efficiency and model performance by increasing the rank with sinusoidal functions without adding parameters. The contributions are a theoretical framework, demonstrations of the performance and evaluation on several tasks such as computer vision, 3D shape modeling and NLP. A large part of the paper is the mathematical analysis of low-rank decomposition and the derivation of a non-linear low-rank decomposition as well as showing mathematically why its rank is larger. The supplementary material holds detailed proofs for the propositions and theorems.

**Strengths:**

- The idea is simple and fascinating. If there are no flaws in the proof, which I didn't find, this has a potential large impact in the domain.
- The mathematical background is well explained, motivated and proven.
- Figures like Figure 2 help in understanding the impact of the method's parameter.
- The method is shown on a variety of very different tasks from classification to Nerf.
- Extreme scenarios are studied like Nerf reconstruction using Lora with rank k=1

**Weaknesses:**

- The method is only compared on self-implemented baselines and seems to underperform compared to other improvements in the domain. E.g. One of the citations even mentions DoRA but as an example of application of Lora techniques. They do have results of LLaMA3-8B LoRA which are similar to the results achieved in this paper but with DoRA they improve the performance on commonsense reasoning beyond this paper. When presenting an improved LoRA training technique the authors should compare their performance not only to the vanilla LoRA case but to other techniques like DoRA and ideally implement and evaluate an improved DoRA although I do not see with this specific work how that would be applied. Nevertheless, it seems important state-of-the-art here is missing, which is needed to position the paper in the research landscape.

There are a lot of minor syntax errors or errors in the equations which need to be corrected and do not give a polished impression (See questions)

**Questions:**

Figure 4 in the text is referenced as table 4.

Why is DoRA cited only once as Lora technique but then there is no reference to it's improved performance?

In line 182, and 765 is the A_{ij} \neq 0 maybe a A_{ij \neq 0} ? It looks as if the indexes should not be 0 and not as if the assumption of A \neq 0 is formalized.

The operator norm in equation 5 has one | too much.

Why is the observation in line 786 important for 11? This should be true in any case.

Line 799 Says Equation equation 13. Probably an issue with cref

The sentence starting in 855 is somehow messed up.

Using parameter counts instead of compression rate in Figure 1 for ViT Classification would improve comparison with other state-of-the-art approaches.

Why does an increased rank improve performance? This point may have been demonstrated by experiments but an intuition could be given. Adding certain noises to the matrix may also increase it's rank without increasing it's parameter count but will probably not lead to improved performance. The link between an increased rank and improved performance could be made clear or hypothesized. Seeing that low-rank adaptation decreases performance may be not sufficient to claim an inverse relationship.

In line 480 I am not sure it is good to speak of overfitting regarding to NeRFs because this sounds as if it's negative when it is the desired behavior.

Figure 10 seems to indicate the frequency parameter has an optimum. How hard is it to choose it? Are the best results retrieved from a grid search or is there any intuition about it's value beforehand?

---

> ### Author Response · Authors · 2024-11-21
> **Response by authors**
>
> We thank the reviewer for their review and greatly appreciate the time and effort they have put in reviewing our paper. Below we will answer each of the reviewers questions in detail. We have also made changes as requested by the reviewer and have uploaded an updated version of our paper.
>
> 1. **Figure 4 in the text is referenced as table 4.**
>
> Thank you we found this error on line 405 and have changed it.
>
> 2. **Why is DoRA cited only once as Lora technique but then there is no reference to it's improved performance?**
>
> We thank the reviewer for pointing out this inconsistency. We have added results for DoRA in appendix A.2.1 Table 7. For the convenience of the reviewer we have given our DoRA results below. As can be seen from the table our sine DoRA outperforms DoRA at each rank level and rank 8 sine DoRA matches the performance of rank 32 DoRA.
>
> **Performance and parameter count of the LLaMA 3-8B model fine-tuned using the DoRA and sine DoRA methods across varying ranks**
>
> | **Method**              | **Params** | **BoolQ** | **PIQA** | **SIQA** | **HS**  | **WG**  | **ARC-e** | **ARC-c** | **OBQA** | **Avg.** |
> |--------------------------|------------|-----------|----------|----------|---------|---------|-----------|-----------|----------|----------|
> | DoRA$_{k=8}$      | 14.9M      | 73.2      | 87.7     | 79.9     | 94.7    | 84.5    | 89.3      | 78.0      | 83.2     | 83.8     |
> | Sine DoRA$_{k=8}$ | 14.9M      | **73.9**  | **89.0** | **81.0** | **95.3**| **86.1**| **90.1**  | **79.0**  | **87.0** | **85.2** |
> | DoRA$_{k=16}$    | 29.1M      | 74.5      | 88.8     | 80.3     | **95.5**| 84.7    | **90.1**  | 79.1      | 87.2     | 85.0     |
> | Sine DoRA$_{k=16}$| 29.1M      | **75.1**  | **89.0** | **81.0** | 95.3    | **86.1**| 90.0      | **79.3**  | **86.2** | **85.3** |
> | DoRA$_{k=32}$    | 57.4M      | 74.6      | **89.3** | 79.9     | 95.5    | 85.6    | **90.5**  | **80.4**  | **85.8** | 85.2     |
> | Sine DoRA$_{k=32}$| 57.4M      | **75.8**  | **89.3** | **80.3** | **95.9**| **86.1**| 90.2      | 79.4      | 85.4     | **85.3** |

---

> ### Author Response · Authors · 2024-11-21
> **Response by authors**
>
> 3. **In line 182, and 765 is the A_{ij} \neq 0 maybe a A_{ij \neq 0} ? It looks as if the indexes should not be 0 and not as if the assumption of A \neq 0 is formalized.**
>
> The way we have written this in the paper is correct. $A_{ij}$ denotes the $(i,j)$ entry of the matrix $A$. In line 182 and 765 the statement $A_{ij} \neq 0$ means the $(i,j)$ entry is not zero. This is used because we want to take the minimum over all the matrix entries of $A$ that are not zero.
>
> 4. **The operator norm in equation 5 has one | too much.**
>
> Thank you. However, the operator norm as written is correct. The notation may look confusing because we are taking the absolute value of the matrix $A$ and then doing a square root. So we first form
> $|A|$ which is the component-wise absolute value of $A$ and then we take the component-wise square root of $|A|$ forming
> $\sqrt{|A|}$ and then take the operator norm of $\sqrt{|A|}$ forming $|| \sqrt{|A|}||_2$. The extra $|$ you see is coming from the absolute value part of $A$. To make this easier on the eyes we have changed the operator norm in equation (5) to be larger so the reader can clearly see what our notation is doing. We hope this make things easier to read.
>
> 5. **Why is the observation in line 786 important for 11? This should be true in any case.**
>
> Thank you. The line 786 is redundant. As you say it is true in any case. We have removed that line.
>
> 6. **Line 799 Says Equation equation 13. Probably an issue with cref**
>
> Thank you! This has been corrected.
>
> 7. **The sentence starting in 855 is somehow messed up.**
>
> Thank you! You are absolutely right. We have removed that sentence.
>
> 8. **Using parameter counts instead of compression rate in Figure 1 for ViT Classification would improve comparison with other state-of-the-art approaches.**
>
>  Thank you! We have changed figure 1 and figure 4 to show parameter counts instead of compression rates.
>
> 9. **Why does an increased rank improve performance? This point may have been demonstrated by experiments but an intuition could be given. Adding certain noises to the matrix may also increase it's rank without increasing it's parameter count but will probably not lead to improved performance......**
>
> High-rank matrices offer greater degrees of freedom than low-rank matrices, enhancing their representational capacity. For complex datasets, we hypothesize that this increased flexibility allows the network to capture and learn key features more effectively. While we acknowledge the reviewer’s suggestion that adding noise can also increase rank, this approach risks propagating noise through the output, potentially reducing the model's effectiveness. Our empirical results across all tasks support our hypothesis, demonstrating that increasing rank using a sine activation provides an effective way to model complex signals with low parameters. This approach balances representational power and parameter efficiency. We have expanded on this in Appendix A.1, Remark 3.
>
> 10. **In line 480 I am not sure it is good to speak of overfitting regarding to NeRFs because this sounds as if it's negative when it is the desired behavior.**
>
> Thank you for the suggestion. We have decided to remove ``overfitting" from that sentence.
>
> 11. **Figure 10 seems to indicate the frequency parameter has an optimum. How hard is it to choose it? Are the best results retrieved from a grid search or is there any intuition about it's value beforehand?**
>
> We chose the frequency by treating it as a hyperparameter and tuning it. In general the best results are indeed found via a grid search. We do however believe that finding a principled way to choose the frequency is an important problem that we plan on taking up for future work.

---

> ### Comment · Reviewer_Gygt · 2024-11-21
>
> I have read the answers of the authors and am particularly pleased with the investigation into Dora. Even though a discussion into the effect size could follow at the conference, I think the paper should be accepted and the method discussed in the larger audience of the ICLR. Due to the numerous improvements and investigations I improved my rating to accept.

---

### Official Review · Reviewer_C1Nj · 2024-11-05

**Soundness:** 3
**Presentation:** 3
**Contribution:** 2
**Rating:** 6
**Confidence:** 4

**Summary:**

This paper presents a theoretical framework that integrates a sinusoidal function within the low rank decomposition process to achieve efficient learning. The proposed method was evaluated on several applications such as LLMs parameter efficient fine-tuning, NeRF and 3D shape modeling.

**Strengths:**

+ The paper provides theoretical justification on benefit of the proposed sine-activated low rank matrices without adding additional parameters.
+ The experimental results show the proposed sine-activated low-rank decomposition works better than the original LoRA.

**Weaknesses:**

- The paper talks about the benefits of using a sine function for increasing rank, but it could explain more about why this function was chosen over others and why it works. A discussion of its advantages would strengthen this work.

- While the proposed method is applied across various domains, the paper lacks a comprehensive comparison with existing low-rank approximation techniques. There are many existing LoRA variants and improved versions, e.g. [a] [b].

[a] LoLDU: Low-Rank Adaptation via Lower-Diag-Upper Decomposition for Parameter-Efficient Fine-Tuning
[b] MELoRA: Mini-Ensemble Low-Rank Adapters for Parameter-Efficient Fine-Tuning

- It would be good to provide space and time complexity analysis of the proposed method.

- There is a lack of quantitative results on a standard dataset for the 3D shape modeling task. It would be good to evaluate the method on other popular tasks such as text to image generation.

- How do we choose the frequency parameter (ω) in practice? Is there an easy way to adjust it? As per the paper, (ω) higher increases the rank. But do we always pick highest (ω) ? How do we effectively use different frequencies within an interval ?

- For each application, the implementation details of the proposed method should be provided.

- Are there any implementation challenges, like increased computation time during training? Does this technique slow down the model
during inference, especially in environments with limited resources?

**Questions:**

Please refer to the weakness section.

---

> ### Author Response · Authors · 2024-11-21
> **Response by authors**
>
> We thank the reviewer for their review and greatly appreciate the time and effort they have put in reviewing our paper. Below we will answer each of the reviewers questions in detail. We have also made changes as requested by the reviewer and have uploaded an updated version of our paper.
>
> 1. **The paper talks about the benefits of using a sine function for increasing rank, but it could explain more about why this function was chosen over others and why it works....**
>
> Our main reason for using a sine function is that it has periodic behaviour around the origin that can be controlled by a frequency parameter. This allows us to show that for any matrix $\mathbf{A}$, that is not the zero matrix, there are frequencies $\omega > 0$ that allow us to lower bound the Frobenius norm of
> $\sin(\omega\mathbf{A})$ by a quantity that depends linearly on the frequency $\omega$ and upper bound the operator norm of
> $\sin(\omega\mathbf{A})$ by a quantity that depends sub-linearly on the frequency $\omega$. This is the content of lemmas 1 and 2 in appendix A.1 and are the two key reasons why we can increase the rank of $\mathbf{A}$ after applying a sine function with a large enough frequency. The proof of both lemmas uses the fact that sine is a periodic function whose frequency can be controlled by a positive parameter $\omega > 0$ which is what we found was critical. In general, other non-linearities such as sigmoid and ReLU do not satisfy such properties and this is why we found that they cannot be made to increase rank. We have added a remark in the appendix after the proof of lemma 2 that clearly explains this for the reader.

---

> ### Author Response · Authors · 2024-11-21
> **Response by authors**
>
> 2. **While the proposed method is applied across various domains, the paper lacks a comprehensive comparison with existing low-rank approximation techniques. There are many existing LoRA variants and improved versions, e.g. [a] [b]....**
>
> We thank the reviewer for this suggestion. We would like to remind the reviewer that ICLR reviewer instructions state that works are considered contemporary if they were published at a peer reviewed venue after 1 July 2024, and as such authors aren't required to directly compare to such works or works that have not been published at a peer reviewed venue (see https://iclr.cc/Conferences/2025/ReviewerGuide last question and answer). We point out to the reviewer that reference [a] has not been published at a peer reviewed venue and was recently uploaded to the ArXiv on 17th of October 2024. Furthermore, [b] was published at ACL on 24th of August 2024. However, we are committed to making our paper as relevant as possible to the wider community. Therefore, as suggested by the other reviewers, we have ran our method on the recent **state-of-the-art** work DoRA [1] which was published on 21 July 2024 in ICML. The results are shown in appendix A.2.1 Table 7.
>
> For the convenience of the reviewer the results for our method on DORA is shown in the table below. As can be seen our sine DoRA outpeforms the standard DoRA at each rank level, and Sine DoRA matches the performance of Rank 32 Dora at only Rank 8.
>
>
>
> **Performance and parameter count of the LLaMA 3-8B model fine-tuned using the DoRA and sine DoRA methods across varying ranks**
>
> | **Method**              | **Params** | **BoolQ** | **PIQA** | **SIQA** | **HS**  | **WG**  | **ARC-e** | **ARC-c** | **OBQA** | **Avg.** |
> |--------------------------|------------|-----------|----------|----------|---------|---------|-----------|-----------|----------|----------|
> | DoRA$_{k=8}$      | 14.9M      | 73.2      | 87.7     | 79.9     | 94.7    | 84.5    | 89.3      | 78.0      | 83.2     | 83.8     |
> | Sine DoRA$_{k=8}$ | 14.9M      | **73.9**  | **89.0** | **81.0** | **95.3**| **86.1**| **90.1**  | **79.0**  | **87.0** | **85.2** |
> | DoRA$_{k=16}$    | 29.1M      | 74.5      | 88.8     | 80.3     | **95.5**| 84.7    | **90.1**  | 79.1      | 87.2     | 85.0     |
> | Sine DoRA$_{k=16}$| 29.1M      | **75.1**  | **89.0** | **81.0** | 95.3    | **86.1**| 90.0      | **79.3**  | **86.2** | **85.3** |
> | DoRA$_{k=32}$    | 57.4M      | 74.6      | **89.3** | 79.9     | 95.5    | 85.6    | **90.5**  | **80.4**  | **85.8** | 85.2     |
> | Sine DoRA$_{k=32}$| 57.4M      | **75.8**  | **89.3** | **80.3** | **95.9**| **86.1**| 90.2      | 79.4      | 85.4     | **85.3** |
>
>
> [1] Liu et al. DoRA: Weight-Decomposed Low-Rank Adaptation. ICML 2024

---

> ### Author Response · Authors · 2024-11-21
> **Response by authors**
>
> 3. **It would be good to provide space and time complexity analysis of the proposed method.**
>
> In the appendix we have added quantitative results on number of parameters, memory and training time see appendix A.2.1. While a theoretical analysis of the space-time complexity of our method is useful we note that adding a sine does not increase any parameters and thus only adds overheads during backpropagation when a derivative of the sine function is computed and during forward propagation when the sine function is applied to the weight matrix. This will be dominated by other components of the optimization process.
>
> 4. **There is a lack of quantitative results on a standard dataset for the 3D shape modeling task. It would be good to evaluate the method on other popular tasks such as text to image generation.**
>
> We carried out experiments in the application of binary occupancy fields for 3D shape reconstruction as we found that the original binary occupancy paper [2] had 3000 citations and was commonly used for 3D shape modeling. Therefore we decided to test our method on this novel task. The work [2] used the Stanford 3D Scanning Repository (https://graphics.stanford.edu/data/3Dscanrep/) as their data sets and we thus followed that approach. To show the reviewer that we are committed to showing that our methodology works for various instances, we have run further experiments on binary occupancy fields using two more instances from the Stanford 3D Scanning Repository. The complete quantitative results can be found in appendix A.2.5. For you convenience we have shown the results below.
>
> We thank the reviewer for suggesting text to image tasks however this is a completely different field to explore for our method. To show that our method is applicable across a variety of fields we carried out tasks on fine tuning LLMs, pre-training vision transformers, neural radiance fields and 3d shape modeling. Furthermore, we have added a new section in the appendix, namely appendix A.2.3, on applying our method to a state of the art convolutional architecture. We believe text to image is an interesting area to consider and will consider it for future work.
>
>
> | **Method**                        | **# Params** | **Thai (IoU)** | **Lucy (IoU)** | **Dragon (IoU)** |
> |-----------------------------------|--------------|----------------|----------------|------------------|
> | Full-Rank                         | 132K         | 97.2           | 97.8           | 98.7             |
> | Low-Rank$_{k=1}$            | 2.8K         | 84.3           | 79.3           | 90.4             |
> | Sine Low-Ran$_{k=1, ω=200}$|      2.8K        | **90.8**       | **90.7**       | **94.6**         |
> | Low-Rank$_{k=2}$            | 3.8K         | 88.0           | 89.4           | 90.9             |
> | Sine Low-Rank$_{k=2, ω=100}$|     3.8K         | **92.0**       | **93.2**       | **96.6**         |
> | Low-Rank$_{k=5}$            | 6.9K         | 93.4           | 94.8           | 96.9             |
> | Sine Low-Rank$_{k=5, ω=50}$ |      6.9K        | **94.3**       | **95.3**       | **97.4**         |
> | Low-Rank$_{k=20}$           | 22.8K        | **95.4**       | 96.2           | 98.0             |
> | Sine Low-Rank$_{k=20, ω=20}$|     22.8K         | **95.4**       | **96.3**       | **98.1**         |
>
> [2]: Occupancy networks: Learning 3d reconstruction in function space, CVPR 2019.

---

> ### Author Response · Authors · 2024-11-21
> **Response by authors**
>
> 5. **How do we choose the frequency parameter (ω) in practice? Is there an easy way to adjust it?....**
>
> We chose the frequency parameter by treating it as a hyperparameter and tuning it. We find that the optimal frequency varies according to the target rank. We believe finding a principled way to choose the frequency is an important problem that we plan on taking up for future work.
>
> 6. **For each application, the implementation details of the proposed method should be provided.**
>
>  We thank the reviewer for this suggestion. We have added all implementation details for each experiment in the appendix.
>
> 7. **Are there any implementation challenges, like increased computation time during training? Does this technique slow down the model during inference, especially in environments with limited resources?**
>
> One of the advantages of our method is that it is a plug-in method that works on all feedforward architectures within a variety of neural network models. This was the main reason we applied it to a variety of different architectures such as ViTs, fine tuning LLMs, NeRFs, Binary Occupancy Fields, to show it works very easily in all these settings. The main challenge we faced is that we had to tune the frequency hyperparameter of the sine function for each experiment. We have uploaded code for the reviewer where it can be seen that our method is a simple plug in method.

---

> > ### Comment · Reviewer_C1Nj · 2024-11-26
> > **Post rebuttal**
> >
> > The authors provided additional experiments and analysis. I will raise my score.

---

> > > ### Author Response · Authors · 2024-11-29
> > >
> > > We thank the reviewer for increasing their score and taking our additional experiments, revised submissions, and response to queries into consideration. Before the discussion period ends we would like to check if there are any remaining concerns we may clarify that would increase your confidence that this paper should be accepted?

---

### Author Response · Authors · 2024-11-21
**Response by authors**

We thank all reviewers for taking the time to read and review our paper. We have responded to each reviewer individually and uploaded an updated version of paper.

---

### Meta-Review · Area_Chair_kFXM · 2024-12-19

**Metareview:**

Motivated by the need of compact and parameter-efficient architectures in machine learning, this paper proposes a novel theoretical framework that integrates a sinusoidal function within the low-rank decomposition process aimed at lowering the number of model parameters with no or only slight degradation in performance in neural network architectures. The goodness of this approach has been  validated in a number of applications, namely Vision Transformers (ViT), Large Language Models (LLMs), Neural Radiance Fields (NeRF) and 3D shape modelling.

This work originally received almost all positive evaluations (5, 6, 8, 6), turned in unanimous appreciation after rebuttal (6, 8, 8, 6).

Positive aspects regard the approach itself, theoretically well presented and justified, as well as validated by a convincing experimental analysis.

Some remarks have also been raised, mainly oriented at requesting to better specify and explain some parts of the approach and experiments, hyper-parameter tuning and asking clarifications for the motivations of some of the methodological choices.

In conclusion, this paper is accepted for publication to ICLR 2024.

**Additional Comments On Reviewer Discussion:**

See above

---

### Decision · Program_Chairs · 2025-01-22

Accept (Poster)